# Gradual Optimization Learning for Conformational Energy Minimization

**Artem Tsypin**[1][✉]**, Leonid Ugadiarov**[2,4]**, Kuzma Khrabrov**[1]**, Alexander Telepov**[1]**,**

**Egor Rumiantsev**[1]**, Alexey Skrynnik**[1,2]**, Aleksandr Panov**[1,2,4]**,**

**Dmitry Vetrov**[5]**, Elena Tutubalina**[1,3,6]**, Artur Kadurin** [1,7][✉]

[1]AIRI, Moscow [2]FRC CSC RAS, Moscow [3]Sber AI, Moscow

[4]MIPT, Dolgoprudny [5]Constructor University, Bremen

[6]ISP RAS Research Center for Trusted Artificial Intelligence, Moscow

[7]Kuban State University, Krasnodar

[✉]{Tsypin, Kadurin}@airi.net

## Abstract

Molecular conformation optimization is crucial to computer-aided drug discovery and materials design. Traditional energy minimization techniques rely on iterative optimization methods that use molecular forces calculated by a physical simulator (oracle) as anti-gradients. However, this is a computationally expensive approach that requires many interactions with a physical simulator. One way to accelerate this procedure is to replace the physical simulator with a neural network. Despite recent progress in neural networks for molecular conformation energy prediction, such models are prone to errors due to distribution shift, leading to inaccurate energy minimization. We find that the quality of energy minimization with neural networks can be improved by providing optimization trajectories as additional training data. Still, obtaining complete optimization trajectories demands a lot of additional computations. To reduce the required additional data, we present the Gradual Optimization Learning Framework (GOLF) for energy minimization with neural networks. The framework consists of an efficient data-collecting scheme and an external optimizer. The external optimizer utilizes gradients from the energy prediction model to generate optimization trajectories, and the data-collecting scheme selects additional training data to be processed by the physical simulator. Our results demonstrate that the neural network trained with GOLF performs *on par* with the oracle on a benchmark of diverse drug-like molecules using significantly less additional data.

## 1 Introduction

Numerical quantum chemistry methods are essential for modern computer-aided drug discovery and materials design pipelines. They are used to predict the physical and chemical properties of candidate structures (Matta & Boyd, 2007; Oglic et al., 2017; Tielker et al., 2021). *Ab initio* property prediction framework for a specific molecule or material could be divided into three main steps as follows: (1) find a low-energy conformation of a given atom system, (2) compute its electron structure with quantum chemistry methods, and (3) calculate properties of interest based on the latest. The computational cost of steps (1) and (2) is defined by the specific physical simulator (oracle $\mathcal{O}$) varying from linear to exponential complexity w.r.t the number of atoms or electrons in the system (Sousa et al., 2007). Overall, the more accurate the oracle is, the more computationally expensive its operations become.

---

[✉]Corresponding authors.

The traditional approach to the problem of obtaining low-energy molecular conformations is to run an iterative optimization process using physical approximations, such as those provided by the Density-functional theory (DFT) methods (Kohn & Sham, 1965), as they are reasonably accurate. However, for large molecules, even a single iteration may take up several hours of CPU-compute (Gilmer et al., 2017). Therefore, it is crucial to develop alternative approaches (such as Neural Network-based) that reduce the computational complexity of iterative optimization.

The recent growth in computational power led to the emergence of molecular databases with computed quantum properties (Ruddigkeit et al., 2012; Ramakrishnan et al., 2014; Isert et al., 2022; Khrabrov et al., 2022; Jain et al., 2013). For example, nablaDFT (Khrabrov et al., 2022) consists of more than $5 \times 10^6$ conformations for around $10^6$ drug-like molecules. This data enabled deep learning research for many molecule-related problems, such as conformational potential energy and quantum properties prediction with **N**eural **N**etwork **P**otentials (NNP) (Chmiela et al., 2017; Schütt et al., 2017; Chmiela et al., 2018; 2020; Schütt et al., 2021; Shuaibi et al., 2021; Gasteiger et al., 2020; 2021; Chmiela et al., 2023), and conformational distribution estimation (Simm & Hernández-Lobato, 2019; Xu et al., 2021; Ganea et al., 2021; Xu et al., 2022; Jing et al., 2022; Shi et al., 2021; Luo et al., 2021). Naturally, there have been several works that utilize deep learning to tackle the problem of obtaining low-energy conformations. One approach is to reformulate this task as a conditional generation task (Guan et al., 2021; Lu et al., 2023); see Section 2 for further details. Another solution is to train an NNP to predict the potential energy of a molecular conformation and use it as a force field for relaxation (Unke et al., 2021). Assuming the NNP accurately predicts the energy, its gradients can be used as interatomic forces (Schütt et al., 2017). Such a technique allows for gradient-based optimization without a physical simulator, significantly reducing computational complexity.

In this work, we aim to improve the training of NNPs for obtaining low-energy conformations. We trained NNPs on the subset of nablaDFT dataset (Khrabrov et al., 2022) and observed that such models suffer from the distribution shift when used in the optimization task (see Figure 1). To alleviate the distribution shift and improve the quality of energy minimization, we enriched the training dataset with optimization trajectories (see Section 4) generated by the oracle. Our experiments demonstrate that it requires more than $5 \times 10^5$ additional oracle interactions to match the quality of a physical simulator (see Table 1). These models trained on enriched datasets are used as baselines for our proposed approach.

In this paper, we propose the GOLF — **G**radual **O**ptimization **L**earning **F**ramework for the training of NNPs to generate low-energy conformations. GOLF consists of three components: (i) a genuine oracle $\mathcal{O}_G$, (ii) an optimizer, and (iii) a surrogate oracle $\mathcal{O}_S$ that is computationally inexpensive. The $\mathcal{O}_G$ is an accurate but computationally expensive method used to calculate ground truth energies and forces, and we consider a setting with a limited budget on $\mathcal{O}_G$ interactions. The optimizer (e.g., Adam (Kingma & Ba, 2014) or L-BFGS (Liu & Nocedal, 1989)) utilizes NNP gradients to produce optimization trajectories. The $\mathcal{O}_S$ determines which conformations are added to the training set. We use Psi4 (Smith et al., 2020), a popular software for DFT-based computations, as the $\mathcal{O}_G$, and RDKit's (Landrum et al., 2022) MMFF (Halgren, 1996) as the $\mathcal{O}_S$. The NNP training cycle consists of three steps. First, we generate a batch of optimization trajectories and evaluate all conformations with $\mathcal{O}_S$. Then we select the first conformation from each trajectory for which the NNP poorly predicts interatomic forces w.r.t. $\mathcal{O}_S$ (see Section 5), calculate its ground truth energy and forces with the $\mathcal{O}_G$, and add it to the training set. Lastly, we update the NNP by training on batches sampled from initial and collected data. We train the model until we exceed the computational budget for additional $\mathcal{O}_G$ interactions. We show (see Section 6.2) that NNPs trained with GOLF on the nablaDFT (Khrabrov et al., 2022) perform on par with $\mathcal{O}_G$ while using 50x less additional data compared to the straightforward approach described in the previous paragraph. We also show similar results on another diverse dataset of drug-like molecules called SPICE (Eastman et al., 2023). We publish[1] the source code for GOLF along with optimization trajectories datasets, training, and evaluation scripts.

Our contributions can be summarized as follows:

- We study the task of conformational optimization and find that NNPs trained on existing datasets are prone to the distribution shift, leading to inaccurate energy minimization.

---

[1] https://github.com/AIRI-Institute/GOLF

- We propose a straightforward approach to deal with the distribution shift by enriching the training dataset with optimization trajectories (see Figure 1). Our experiments show that additional $5 \times 10^5$ conformations make the NNP perform comparably with the DFT-based oracle $\mathcal{O}_G$ on the task of conformational optimization.

- We propose a novel framework (GOLF) for data-efficient training of NNPs, which includes a data-collecting scheme along with an external optimizer. We show that models trained with GOLF perform on par with the physical simulator on the task of conformational optimization using 50x less additional data than the straightforward approach.

## 2 RELATED WORK

**Conformation generation**    Several recent papers have proposed different approaches for predicting molecule's 3D conformers. Xu et al. (2021) utilize normalizing flows to predict pairwise distances between atoms for a given molecular structure with subsequent relaxation of the generated conformation. Ganea et al. (2021) construct the molecular conformation by iteratively assembling it from smaller substructures. Xu et al. (2022); Wu et al. (2022); Jing et al. (2022); Huang et al. (2023); Fan et al. (2023) address the conformational generation task with diffusion models (Sohl-Dickstein et al., 2015). Other works employ variational approximations (Zhu et al., 2022; Swanson et al., 2023), and Markov Random Fields (Wang et al., 2022). We evaluate these approaches in Section 6.1. Despite showing promising geometrical metrics, such as Root-mean-square deviation of atomic positions (RMSD), on the tasks reported in the various papers, these models perform poorly in terms of geometry and potential energy on the optimization task. In most cases, additional optimization with a physical simulator is necessary to get a valid conformation.

**Geometry optimization**    Guan et al. (2021); Lu et al. (2023) frame the conformation optimization problem as a conditional generation task and train the model to generate low-energy conformations conditioned on RDKit-generated (or the randomly sampled from the pseudo optimization trajectory) conformations by minimizing the RMSD between the corresponding atom coordinates. As RMSD may not be an ideal objective for the conformation optimization task (see Section 6.1), we focus on accurately predicting the interatomic forces along the optimization trajectories in our work.

**Additional oracle interactions**    Zhang et al. (2018) show that additional data from the oracle may increase the energy prediction precision of NNP models. Following this idea, Kulichenko et al. (2023) propose an active learning approach based on the uncertainty of the energy prediction to reduce the number of additional oracle interactions. The main limitation of this approach is that it requires training a separate NNP ensemble for every single molecule. Chan et al. (2019) parametrize the molecule as a set of rotatable bonds and utilize the Bayesian Optimization with Gaussian Process prior to efficiently search for low-energy conformations. However, this method requires using the oracle during the inference, which limits its applications. The OC2022 (Tran* et al., 2022) provides relaxation trajectories for catalyst-adsorbate pairs. However, no in-depth analysis of the effects of such additional data on the quality of optimization with NNPs is provided.

To sum up, we believe it necessary to explore further the ability of NNPs to optimize molecular conformations according to their energy. Our experiments (see Section 6) show that additional oracle information significantly increases the optimization quality. Since this information may be expensive, we aim to reduce the number of additional interactions while maintaining the quality *on par* with the oracle.

## 3 NOTATION AND PRELIMINARIES

We define the conformation $s = \{z, X\}$ of the molecule as a pair of atomic numbers $z = \{z_1, \ldots, z_n\}, z_i \in \mathbb{N}$ and atomic coordinates $X = \{x_1, \ldots, x_n\}, x_i \in \mathbb{R}^3$, where $n$ is the number of atoms in the molecule. We define the oracle $\mathcal{O}$ as a function that takes conformation $s$ as an input and outputs its potential energy $E_s^{\text{oracle}} \in \mathbb{R}$ and interatomic forces $F_s^{\text{oracle}} \in \mathbb{R}^{n \times 3}$: $E_s^{\text{oracle}}, F_s^{\text{oracle}} = \mathcal{O}(s)$. To denote the ground truth interatomic force acting on the $i$-th atom, we use $F_{s,i}^{\text{oracle}}$. We use different superscripts to denote energies and forces calculated by different physical simulators. For

example, we denote the RDKit's MMFF-calculated energy as $E_s^{\text{MMFF}}$ and the Psi4-calculated energy as $E_s^{\text{DFT}}$.

We denote the NNP for the prediction of the potential energy of the conformation parametrized by weights $\boldsymbol{\theta}$ as $f(s; \boldsymbol{\theta}) : \{\boldsymbol{z}, \boldsymbol{X}\} \to \mathbb{R}$. Following (Schütt et al., 2017; Schütt et al., 2021), we derive forces from the predicted energies:

$$\boldsymbol{F}_i(s; \boldsymbol{\theta}) = -\frac{\partial f(s; \boldsymbol{\theta})}{\partial \boldsymbol{x}_i}, \tag{1}$$

where $\boldsymbol{F}_i \in \mathbb{R}^3$ is the force acting on the $i$-th atom as predicted by the NNP. We follow the standard procedure (Schütt et al., 2017; Schütt et al., 2021; Gasteiger et al., 2020; Musaelian et al., 2022) and train the NNP to minimize the MSE between predicted and ground truth energies and forces:

$$\mathcal{L}(s, E_s^{\text{oracle}}, \boldsymbol{F}_s^{\text{oracle}}; \boldsymbol{\theta}) = \rho \|E_s^{\text{oracle}} - f(s; \boldsymbol{\theta})\|^2 + \frac{1}{n} \sum_{i=1}^{n} \left\| F_{i,s}^{\text{oracle}} - \boldsymbol{F}_i(s; \boldsymbol{\theta}) \right\|^2, \tag{2}$$

where $\mathcal{L}(s, E_s^{\text{oracle}}, \boldsymbol{F}_s^{\text{oracle}}; \boldsymbol{\theta})$ is the loss function for a single conformation $s$, and $\rho$ is the hyperparameter accounting for different scales of energy and forces.

To collect the ground truth optimization trajectories (see Section 4), we use the OPTIMIZE method from Psi-4 and run optimization until convergence. Optimizer **Opt** (L-BFGS, Adam, SGD-momentum) utilizes the forces $\boldsymbol{F}(s; \boldsymbol{\theta}) \in \mathbb{R}^{n \times 3}$ to get NNP-optimization trajectories $s_0, \ldots, s_T$, where $s_0$ is the initial conformation:

$$s_{t+1} = s_t + \alpha \mathbf{Opt}(\boldsymbol{F}(s_t; \boldsymbol{\theta})). \tag{3}$$

Here, $\alpha$ is the optimization rate hyperparameter, and $T$ is the total number of NNP optimization steps.

In this work, we use NNPs trained on different data. To train the baseline model $f^{\text{baseline}}(\cdot; \boldsymbol{\theta})$, we use the fixed subset of nablaDFT $\mathcal{D}_0$ (see Appendix D for more details). It consists of approximately 10000 triplets of the form $\{s, E_s^{\text{DFT}}, \boldsymbol{F}_s^{\text{DFT}}\}$. The $\mathcal{D}_0$ can be extended with the ground truth optimization trajectories obtained with Psi-4 to get datasets denoted according to the total number of additional conformations: $\mathcal{D}_{\text{traj-10k}}, \mathcal{D}_{\text{traj-100k}}$, and so on. The resulting NNPs are dubbed $f^{\text{traj-1k}}(\cdot; \boldsymbol{\theta})$, $f^{\text{traj-10k}}(\cdot; \boldsymbol{\theta})$, and so on respectively. We call the models trained with GOLF (see Section 5) $f^{\text{GOLF-1k}}(\cdot; \boldsymbol{\theta})$, $f^{\text{GOLF-10k}}(\cdot; \boldsymbol{\theta})$, etc.

To evaluate the quality of optimization with NNPs, we use a fixed subset of the nablaDFT dataset $\mathcal{D}_{\text{test}}$, that shares no molecules with $\mathcal{D}_0$. For each conformation $s \in \mathcal{D}_{\text{test}}$ we perform the optimization with the $\mathcal{O}_G$ to get the ground truth optimal conformation $s_{\mathbf{opt}}$ and its energy $E_{s_{\mathbf{opt}}}^{\text{DFT}}$. The quality of the NNP-optimization for $s_t \in s_0, \ldots, s_T$ is evaluated with the percentage of minimized energy:

$$\text{pct}(s_t) = 100\% * \frac{E_{s_0}^{\text{DFT}} - E_{s_t}^{\text{DFT}}}{E_{s_0}^{\text{DFT}} - E_{s_{\mathbf{opt}}}^{\text{DFT}}}. \tag{4}$$

By aggregating $\text{pct}(s_t)$ over $s \in \mathcal{D}_{\text{test}}$, we get the average percentage of minimized energy at step $t$:

$$\overline{\text{pct}}_t = \frac{1}{|\mathcal{D}_{\text{test}}|} \sum_{s \in \mathcal{D}_{\text{test}}} \text{pct}(s_t); \tag{5}$$

Another metric is the residual energy in state $s_t$: $E^{\text{res}}(s_t)$. It is calculated as the delta between $E_{s_t}^{\text{DFT}}$ and the optimal energy:

$$E^{\text{res}}(s_t) = E_{s_t}^{\text{DFT}} - E_{s_{\mathbf{opt}}}^{\text{DFT}}; \tag{6}$$

Similar to $\overline{\text{pct}}_t$, this metric can also be aggregated over the evaluation dataset:

$$\overline{E^{\text{res}}}_t = \frac{1}{|\mathcal{D}_{\text{test}}|} \sum_{s \in \mathcal{D}_{\text{test}}} E^{\text{res}}(s_t). \tag{7}$$

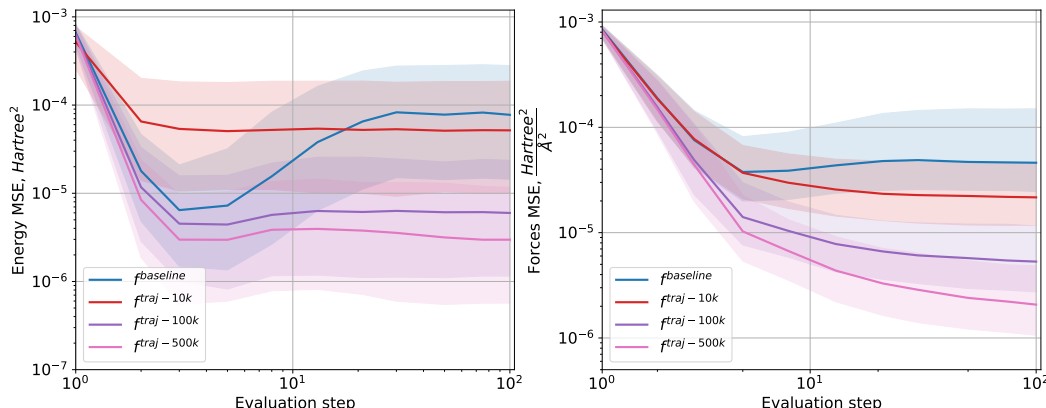

Figure 1: Mean squared error (MSE) of energy and forces prediction for NNPs trained on $\mathcal{D}_0, \mathcal{D}_{\text{traj-10k}}, \mathcal{D}_{\text{traj-100k}}, \mathcal{D}_{\text{traj-500k}}$. To compute the MSE, we collect NNP-optimization trajectories of length $T = 100$ and calculate the ground truth energies and forces on steps $t = 1, 2, 3, 5, 8, 13, 21, 30, 50, 75, 100$. Solid lines indicate the median MSE, and the shaded regions indicate the 10th and the 90th percentiles. Both the x-axis and y-axis are log scaled

Generally accepted chemical precision is 1 kcal/mol (Helgaker et al., 2004). Thus, another important metric is the percentage of conformations for which the residual energy is less than chemical precision. We consider optimizations with such residual energies successful:

$$\text{pct}_{\text{success}} = \frac{1}{|\mathcal{D}_{\text{test}}|} \sum_{s \in \mathcal{D}_{\text{test}}} I \left[ E^{\text{res}}(s_T) < 1 \right]. \tag{8}$$

## 4 CONFORMATION OPTIMIZATION WITH NEURAL NETWORKS

Energy prediction models such as SchNet, DimeNet, and PaiNN can achieve near-perfect quality on tasks of energy and interatomic forces prediction when trained on the datasets of molecular conformations (Schütt et al., 2017; Gasteiger et al., 2020; Schütt et al., 2021; Ying et al., 2021; Shuaibi et al., 2021; Gasteiger et al., 2021; Batzner et al., 2022; Musaelian et al., 2022). In theory, the gradients of these models can be utilized by an external optimizer to perform conformational optimization, replacing the computationally expensive physical simulator. However, in our experiments (see Section 6), this scheme often leads to suboptimal performance in terms of the potential energy of the resulting conformations. We attribute this effect to the distribution shift that naturally occurs during the optimization: As most existing datasets (Isert et al., 2022; Khrabrov et al., 2022; Eastman et al., 2023; Nakata & Maeda, 2023) do not contain conformations sampled from optimization trajectories, the accuracy of prediction deteriorates as the conformation changes along the optimization process. The lack of such conformations in the training can result in either divergence (initial potential energy is lower than the final potential energy) of the optimization or convergence to a conformation with higher final potential energy than the optimization with the oracle.

To alleviate the distribution shift's effect, we propose enriching the training dataset for NNPs with the ground truth optimization trajectories obtained from the $\mathcal{O}_G$. To illustrate the effectiveness of our approach, we conduct a series of experiments. First, we train a baseline model $f^{\text{baseline}}(\cdot; \boldsymbol{\theta})$ on a fixed subset $\mathcal{D}_0$ of small molecules from the nablaDFT dataset. The $\mathcal{D}_0$ ($|\mathcal{D}_0| \approx 10000$) contains conformations for 4 000 molecules, with sizes ranging from 17 to 35 atoms, and the average size of 32.6. Then we train NNPs $f^{\text{traj-}}(\cdot; \boldsymbol{\theta})$ on enriched datasets $\mathcal{D}_{\text{traj-10k}}, \mathcal{D}_{\text{traj-100k}}, \mathcal{D}_{\text{traj-500k}}$, containing approximately $10^4, 10^5, 5 \times 10^5$ additional conformations respectively. The additional data consists of ground truth optimization trajectories obtained from the $\mathcal{O}_G$. Then, we evaluate the NNPs by performing the NNP-optimization on all conformations in $\mathcal{D}_{\text{test}}$ ($|\mathcal{D}_{\text{test}}| \approx 20000$, contains $\approx 10\,000$ molecules) and calculating the MSE between ground truth and predicted energies and forces. We use the L-BFGS as **Opt** due to its superior performance compared to other optimizers (see Appendix B). We run the optimization with an NNP for a fixed number of steps $T = 100$ as we observe that this number is sufficient for the optimization to converge (see Figure 3). Figure 1

Table 1: Optimization metrics for NNPs trained on enriched datasets

| NNP | $f^{\text{baseline}}$ | $f^{\text{traj-10k}}$ | $f^{\text{traj-100k}}$ | $f^{\text{traj-500k}}$ |
|---|---|---|---|---|
| $\overline{\text{pct}}_T(\%) \uparrow$ | $77.9 \pm 21.3$ | $95.1 \pm 7.6$ | $96.2 \pm 8.6$ | $\mathbf{98.8 \pm 7.6}$ |
| $\overline{E^{\text{res}}}_T(\text{kcal/mol}) \downarrow$ | 8.6 | 2.0 | 1.5 | **0.5** |
| $\text{pct}_{\text{success}}(\%) \uparrow$ | 8.2 | 37.0 | 52.7 | **73.4** |

illustrates the effect of the distribution shift on $f^{\text{baseline}}(\cdot; \boldsymbol{\theta})$ (the prediction error increases as the optimization progresses) and its gradual alleviation with the addition of new training data.

Table 1 presents optimization metrics $\overline{\text{pct}}_T, \overline{E^{\text{res}}}_T, \text{pct}_{\text{success}}$ for $T = 100$. Note that the potential energy surfaces of molecules often contain a large number of local minimas (Tsai & Jordan, 1993). Due to this fact and the noise in the predicted forces, the NNP-optimization can converge to a better local minimum than the $\mathcal{O}_G$, resulting in the optimization percentage greater than a hundred: $\text{pct}(s_T) > 100\%$ (see Appendix H for examples). This explains the range of values in Table 1 and the violin plots in Figure 2. We say that the NNP matches the optimization quality of $\mathcal{O}_G$ if its average residual energy $\overline{E^{\text{res}}}_T$ is less than the chemical precision. Table 1 shows that it takes approximately $5 \times 10^5$ additional oracle interactions to match the optimization quality of the $\mathcal{O}_G$. However, it takes on average 590 CPU-seconds to perform a single DFT calculation for a conformation from $\mathcal{D}_0$ with the $\omega$B97X-D/def2-SVP level of theory on our cluster with a total of 960 Intel(R) Xeon(R) Gold 2.60Hz CPU-cores (assuming there are 240 parallel workers each using four threads). This amounts to approximately 9.36 CPU-years of compute for $5 \times 10^5$ additional conformations.

## 5 GOLF

Motivated by the desire to reduce the amount of additional data (and compute) required to match the optimization quality of the $\mathcal{O}_G$, we propose the GOLF. Following the idea of Active Learning, we want to enrich the training dataset with conformations where the NNP's prediction quality deteriorates. We propose to select such conformations by identifying pairs of consecutive conformations $s_t, s_{t+1}$ in NNP-optimization trajectories, for which the potential energy does not decrease: $E_{s_t}^{\text{DFT}} < E_{s_{t+1}}^{\text{DFT}}$. This type of error indicates that the NNP poorly predicts forces in $s_t$, so we add this conformation to the training dataset.

---

**Algorithm 1** GOLF

**Require:** training dataset $\mathcal{D}_0$, genuine oracle $\mathcal{O}_G$, surrogate oracle $\mathcal{O}_S$, optimizer **Opt**, optimization rate $\alpha$, NNP $f(\cdot; \boldsymbol{\theta})$, number of additional $\mathcal{O}_G$ interactions $K$, timelimit $T$, update-to-data ratio $U$

1: Initialize the NNP $f(\cdot; \boldsymbol{\theta})$ with the weights of the baseline NNP model
2: Set $\mathcal{D} \leftarrow Copy(\mathcal{D}_0)$, set $t \leftarrow 0$
3: Sample $s \sim \mathcal{D}$, and calculate its energy with $\mathcal{O}_S : E_{prev} \leftarrow E_s^{\text{MMFF}}$
4: **repeat**
5: $\quad s' \leftarrow s + \alpha \mathbf{Opt}(\boldsymbol{F}(s; \boldsymbol{\theta}))$        ▷ Get next conformation using NNP
6: $\quad$ Calculate new energy with the $\mathcal{O}_S : E_{cur} \leftarrow E_{s'}^{\text{MMFF}}$
7: $\quad$ **if** $E_{cur} > E_{prev}$ **or** $t \geq T$ **then**    ▷ Incorrect forces predicted in $s$, or $T$ reached
8: $\quad\quad$ Calculate $E_s^{\text{DFT}}, \boldsymbol{F}_s^{\text{DFT}} = \mathcal{O}_G(s)$
9: $\quad\quad \mathcal{D} \xleftarrow{\text{add}} \{s, E_s^{\text{DFT}}, \boldsymbol{F}_s^{\text{DFT}}\}$        ▷ Add new data to $\mathcal{D}$
10: $\quad\quad$ Train $f(\cdot; \boldsymbol{\theta})$ on $\mathcal{D}$ using Eq. 2 $U$ times
11: $\quad\quad$ Set $t \leftarrow 0$
12: $\quad\quad$ Sample $s \sim \mathcal{D}$, and calculate its energy with $\mathcal{O}_S : E_{prev} \leftarrow E_s^{\text{MMFF}}$
13: $\quad$ **else**
14: $\quad\quad s \leftarrow s'$
15: $\quad\quad E_{prev} \leftarrow E_{cur}$
16: $\quad\quad t \leftarrow t + 1$
17: $\quad$ **end if**
18: **until** $|\mathcal{D}| - |\mathcal{D}_0| < K$

---

However, this scheme requires estimating the energy for all conformations in generated NNP-optimization trajectories, which makes it computationally intractable. To cope with that, we employ a computationally inexpensive surrogate oracle $\mathcal{O}_S$ to determine which conformations to evaluate with the $\mathcal{O}_G$ and add to the training set. Although the energy estimation provided by the $\mathcal{O}_S$ is less accurate, such simplification allows us to efficiently collect the additional training data and successfully train the NNPs. We chose the RDKit's (Landrum et al., 2022) MMFF (Halgren, 1996) as the $\mathcal{O}_S$ due to its efficiency. In our experiments, it takes 120 microseconds on average on a single CPU core to evaluate a single conformation with MMFF, which is about $5 \times 10^6$ times faster than the average DFT calculation time.

Algorithm 1 describes the GOLF training procedure. We start with an NNP $f(\cdot; \boldsymbol{\theta})$ pretrained on the $\mathcal{D}_0$. We calculate a new optimization trajectory on every iteration using forces from the current NNP and choose a conformation from this trajectory to extend the training set. Then, we update the NNP on batches sampled from the extended training set $\mathcal{D}$. This approach helps the NNP learn the conformational space by gradually descending towards minimal conformations.

## 6    EXPERIMENTS

We evaluate NNPs and baseline models on a subset of nablaDFT $\mathcal{D}_{\text{test}}$, $|\mathcal{D}_{\text{test}}| = 19477$, containing conformations for 10273 molecules. The evaluation dataset $\mathcal{D}_{\text{test}}$ shares no molecules with either $\mathcal{D}_0$ or additional training data. We use PaiNN (Schütt et al., 2021) for all NNP experiments. First, we train a baseline NNP $f^{\text{baseline}}(\cdot; \boldsymbol{\theta})$ on $\mathcal{D}_0$ for $5 \times 10^5$ training steps. To train $f^{\text{traj-}}(\cdot; \boldsymbol{\theta})$ we first initialize the weights of the network with $f^{\text{baseline}}(\cdot; \boldsymbol{\theta})$ and then train it on the corresponding dataset $(\mathcal{D}_{\text{traj-10k}}, \mathcal{D}_{\text{traj-100k}}, \mathcal{D}_{\text{traj-500k}})$ concatenated with $\mathcal{D}_0$ for additional $5 \times 10^5$ training steps. The only exception is the $f^{\text{traj-500k}}(\cdot; \boldsymbol{\theta})$, which is trained for $10^6$ training steps due to a larger dataset.

To train the $f^{\text{GOLF-}}(\cdot; \boldsymbol{\theta})$ models, we select the total number of additional $\mathcal{O}_G$ interactions $K$ and adjust the update-to-data ratio $U$ to keep the total number of updates equal to $5 \times 10^5$. For example, if $K$ is set to $10^4$, we perform $U = 50$ updates for each additional conformation collected (see line 10 of Algorithm 1). The Algorithm 1 describes a non-parallel version of GOLF with a single $\mathcal{O}_G$ . To parallelize the $\mathcal{O}_G$ calculations (line 8), we use a batched version of the Algorithm 1, where a batch of NNP-optimization trajectories is generated and then processed by a large number of parallel DFT oracles.

To evaluate NNPs, we use them to generate optimization trajectories $s_0, \ldots, s_T$, $T = 100$ for all $s \in \mathcal{D}_{\text{test}}$. We then calculate $E^{\text{DFT}}$ at steps $t = \{1, 2, 3, 5, 8, 13, 21, 30, 50, 75, 100\}$, as calculating in every step is computationally expensive. Having calculated $E_{s_T}^{\text{DFT}}$ for all $s \in \mathcal{D}_{\text{test}}$, we can compute $\text{pct}(s_T), E^{\text{res}}(s_t), s \in \mathcal{D}_{\text{test}}$ along with $\overline{\text{pct}}_t, \overline{E^{\text{res}}}_t, \text{pct}_{\text{success}}$. In all our experiments, we use the L-BFGS as **Opt**, except for Appendix B, where we test the effect of different external optimizers on the model's performance. We run the optimization with an NNP for a fixed number of steps $T = 100$ as we observe that this number is sufficient for the optimization to converge (see Figure 3). We report the optimization quality of RDKit's MMFF as a non-neural baseline. If $E_{s_T}^{\text{DFT}} > E_{s_0}^{\text{DFT}}$, we say that the optimization has diverged and do not take such conformations into account when computing $\overline{\text{pct}}_t, \overline{E^{\text{res}}}_t, \text{pct}_{\text{success}}$. We denote the percentage of diverged optimizations as $\text{pct}_{\text{div}}$. We also report well-known metrics COV and MAT (Xu et al., 2021). More information on these metrics can be found in Appendix F. We present all metrics in Table 2.

### 6.1    GENERATIVE BASELINES

To compare our approach with other NN-based methods, we adapt ConfOpt (Guan et al., 2021), Torsional diffusion (TD) (Jing et al., 2022), and Uni-Mol+ (Lu et al., 2023) for the task of conformational optimization. The training dataset is composed of a single conformation for each of 4000 molecules in $\mathcal{D}_0$. We first optimize geometry for each conformation with $\mathcal{O}_G$ and then train the generative models to map initial conformations to final conformations from corresponding optimization trajectories. Table 2 reports the best metrics for each model type. Refer to Appendix G for an in-depth discussion of results. The training details and metrics for all the variants of the models are also reported in Appendix G.

Table 2: Optimization and recall-based metrics. We set $\delta = 0.5$Å when computing the COV. We use **bold** for the best value in each column.

| Methods | $\overline{\mathrm{pct}}_T(\%)\uparrow$ | $\mathrm{pct}_{\mathrm{div}}(\%)\downarrow$ | $\overline{E^{\mathrm{res}}}_{T\,(\mathrm{kc/mol})}\downarrow$ | $\mathrm{pct}_{\mathrm{success}}\ (\%)\uparrow$ | COV(%)↑ | MAT (Å)↓ |
|---|---|---|---|---|---|---|
| RDKit | $85.5 \pm 8.8$ | **0.6** | 5.5 | 4.1 | 54.9 | 0.61 |
| TD | $23.8 \pm 19.8$ | 61.4 | 33.8 | 0.0 | 10.0 | 1.42 |
| ConfOpt | $39.1 \pm 22.8$ | 71.1 | 27.9 | 0.2 | 25.0 | 1.13 |
| Uni-Mol+ | $54.6 \pm 20.4$ | 8.1 | 18.6 | 0.2 | 56.3 | 0.53 |
| $f^{\mathrm{baseline}}$ | $77.9 \pm 21.3$ | 7.5 | 8.6 | 8.2 | 58.8 | 0.55 |
| $f^{\mathrm{rdkit}}$ | $93.0 \pm 11.6$ | 4.4 | 2.8 | 35.4 | 63.8 | 0.51 |
| $f^{\mathrm{traj\text{-}10k}}$ | $95.1 \pm 7.6$ | 4.5 | 2.0 | 37.0 | 63.3 | 0.52 |
| $f^{\mathrm{traj\text{-}100k}}$ | $96.2 \pm 8.6$ | 2.8 | 1.5 | 52.7 | 65.6 | 0.49 |
| $f^{\mathrm{traj\text{-}500k}}$ | $\mathbf{98.8 \pm 7.6}$ | 2.0 | **0.5** | 73.4 | 67.0 | 0.48 |
| $f^{\mathrm{GOLF\text{-}1k}}$ | $97.3 \pm 5.1$ | 3.9 | 1.1 | 62.9 | 71.0 | **0.42** |
| $f^{\mathrm{GOLF\text{-}10k}}$ | $\mathbf{98.8 \pm 5.0}$ | 3.0 | **0.5** | **77.3** | **71.2** | **0.42** |

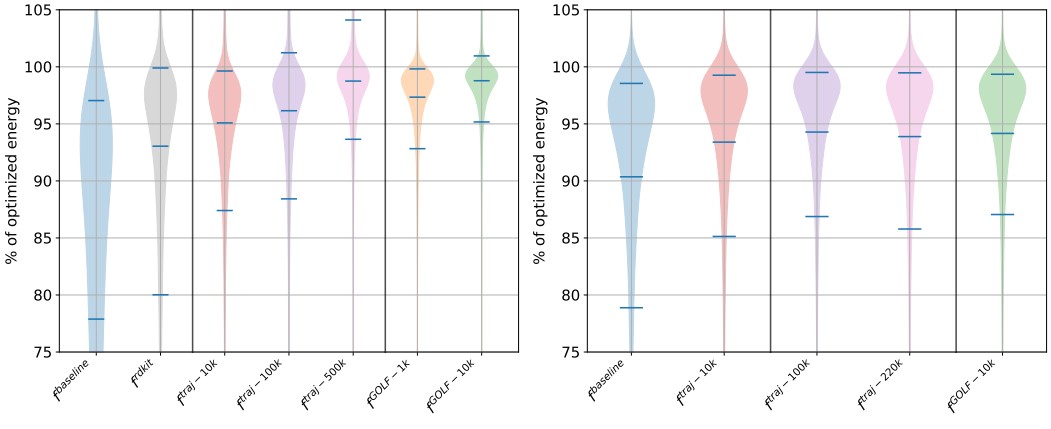

(a) Distribution of $\mathrm{pct}(s_T)$ for NNPs on nablaDFT

(b) Distribution of $\mathrm{pct}(s_T)$ for NNPs on SPICE

Figure 2: Violin plots of the percentage of optimized energy $\mathrm{pct}(s_T)$ calculated for various NNPs on $\mathcal{D}_{\mathrm{test}}$ and $\mathcal{D}_{\mathrm{test}}^{\mathrm{SPICE}}$. Blue marks denote the mean percentage of optimized energy $\overline{\mathrm{pct}}_T$, the 10th, and the 90th quantile.

## 6.2 NNPs TRAINED ON NABLADFT DATASET

To illustrate the performance of various NNPs trained on molecules from the nablaDFT dataset (Khrabrov et al., 2022), we plot the distribution of $\mathrm{pct}(s_T)$ using a violin plot (see Figure 2a). To highlight the data efficiency of the proposed GOLF framework, we report $f^{\mathrm{GOLF\text{-}1k}}(\cdot; \boldsymbol{\theta})$, as well as our primary model $f^{\mathrm{GOLF\text{-}10k}}(\cdot; \boldsymbol{\theta})$. To demonstrate the significance of our proposed data-collecting scheme, we compare the NNPs trained with GOLF against an NNP trained on $\mathcal{D}_{\mathrm{rdkit}} = \{s_{\mathbf{Opt}}^{\mathrm{MMFF}}\}_{s \in \mathcal{D}_0}$, which is composed of the optimal conformations obtained by the $\mathcal{O}_S$ .

As shown in Figure 2a and in Table 2, the NNPs benefit from additional training data and outperform the baseline in terms of all optimization metrics. The $\overline{\mathrm{pct}}_T$ and $\mathrm{pct}_{\mathrm{success}}$ gradually increase with the amount of additional training data both for $f^{\mathrm{traj\text{-}}}(\cdot; \boldsymbol{\theta})$ and $f^{\mathrm{GOLF\text{-}}}(\cdot; \boldsymbol{\theta})$ models. However, the NNPs trained with GOLF require significantly less additional training data: $f^{\mathrm{GOLF\text{-}1k}}(\cdot; \boldsymbol{\theta})$ outperforms $f^{\mathrm{traj\text{-}100k}}(\cdot; \boldsymbol{\theta})$, while using 100 times less data; our main model, $f^{\mathrm{GOLF\text{-}10k}}(\cdot; \boldsymbol{\theta})$ outperforms $f^{\mathrm{traj\text{-}500k}}(\cdot; \boldsymbol{\theta})$ in terms of $\mathrm{pct}_{\mathrm{success}}$, while using 50 times less data. NNPs trained with GOLF also outperform $f^{\mathrm{rdkit}}(\cdot; \boldsymbol{\theta})$, which shows the importance of enriching the dataset with conformations based on the proposed Active Learning-inspired data collecting scheme.

## 6.3 NNPs TRAINED ON SPICE DATASET

To demonstrate the generalization ability of our approach, we perform a similar set of experiments on another diverse dataset of small molecules called SPICE (Eastman et al., 2023). Namely, we select a subset $\mathcal{D}_0^{\text{SPICE}}$ (see Appendix E for detailed description) from the SPICE dataset to be roughly the same size as $\mathcal{D}_0$ and trained a baseline model $f_{\text{SPICE}}^{\text{baseline}}(\cdot; \boldsymbol{\theta})$. We then use the same DFT-based oracle $\mathcal{O}_G$ to get ground truth optimization trajectories and obtain enriched training datasets $\mathcal{D}_{\text{traj-10k}}^{\text{SPICE}}, \mathcal{D}_{\text{traj-100k}}^{\text{SPICE}}, \mathcal{D}_{\text{traj-220k}}^{\text{SPICE}}$. Finally, we train $f_{\text{SPICE}}^{\text{traj-}}(\cdot; \boldsymbol{\theta})$ models and $f_{\text{SPICE}}^{\text{GOLF-10k}}(\cdot; \boldsymbol{\theta})$ model. All the models are evaluated on $\mathcal{D}_{\text{test}}^{\text{SPICE}}$ dataset ($|\mathcal{D}_{\text{test}}^{\text{SPICE}}| = 17724$) that shares no molecules with $\mathcal{D}_0^{\text{SPICE}}$. The results are in Figure 2b and Table 3. It should be noted that the hyperparameters used in these experiments were not specifically optimized for the SPICE dataset, suggesting potential for further improvements in the metrics with tailored adjustments.

Table 3: Optimization metrics for NNPs trained on $\mathcal{D}_0^{\text{SPICE}}$

| NNP | $f^{\text{baseline}}$ | $f^{\text{traj-10k}}$ | $f^{\text{traj-100k}}$ | $f^{\text{traj-220k}}$ | $f^{\text{GOLF-10k}}$ |
|---|---|---|---|---|---|
| $\overline{\text{pct}}_T(\%) \uparrow$ | $90.4 \pm 12.0$ | $93.4 \pm 10.0$ | $\mathbf{94.3 \pm 9.4}$ | $93.9 \pm 9.6$ | $94.2 \pm 8.9$ |
| $\text{pct}_{\text{div}}(\%) \downarrow$ | 4.7 | 6.8 | $\mathbf{2.4}$ | $\mathbf{2.4}$ | 3.2 |
| $\overline{E^{\text{res}}}_T(\text{kcal/mol}) \downarrow$ | 3.6 | 2.4 | $\mathbf{2.1}$ | 2.3 | $\mathbf{2.1}$ |
| $\text{pct}_{\text{success}}(\%) \uparrow$ | 19.7 | 37.4 | $\mathbf{44.2}$ | 41.6 | 40.9 |

## 6.4 LARGE MOLECULES

Finally, we test the ability of our models to generalize to unseen molecules of bigger size. To do that, we collect a dataset $\mathcal{D}_{\text{LM}}$ (LM for **L**arge **M**olecules) of 2000 molecules from the nablaDFT dataset. Sizes of molecules in $\mathcal{D}_{\text{LM}}$ range from 36 atoms to 57 atoms with an average size of 41.8 atoms.

Table 4: Optimization metrics for NNPs trained on $\mathcal{D}_0$

| NNP | $f^{\text{baseline}}$ | $f^{\text{traj-500k}}$ | $f^{\text{GOLF-10k}}$ |
|---|---|---|---|
| $\overline{\text{pct}}_T(\%) \uparrow$ | $77.7 \pm 19.7$ | $97.4 \pm 6.7$ | $\mathbf{97.7 \pm 4.1}$ |
| $\text{pct}_{\text{div}}(\%) \downarrow$ | 5.1 | $\mathbf{1.9}$ | 2.7 |
| $\overline{E^{\text{res}}}_T(\text{kcal/mol}) \downarrow$ | 9.6 | 1.1 | $\mathbf{1.0}$ |
| $\text{pct}_{\text{success}}(\%) \uparrow$ | 4.8 | 58.2 | $\mathbf{61.4}$ |

As it can be seen in Table 4, the $f^{\text{GOLF-10k}}(\cdot; \boldsymbol{\theta})$ matches the quality of ground truth optimization ($\overline{E^{\text{res}}}_T < 1$), the only downside being a lower $\text{pct}_{\text{success}}$ compared to results in Table 2. We hypothesize that this percentage can be increased by adding a small amount of larger molecules to $\mathcal{D}_0$ but leave this for future work.

## 7 CONCLUSION

In this work, we have presented a new framework called GOLF for molecular conformation optimization learning. We show that additional information from the physical simulator can help NNPs overcome the distribution shift and increase their quality on energy prediction and optimization tasks. We thoroughly compare our approach with several baselines, including recent conformation generation models and an inexpensive physical simulator. Using GOLF, we achieve state-of-the-art performance on the optimization task while reducing the number of additional interactions with the physical simulator by a factor of 50 compared to the naive approach. The resulting model matches the DFT methods' optimization quality on a diverse set of drug-like molecules. In addition, we find that our models generalize to bigger molecules unseen during training. We consider the following two directions for future work. First, we plan to adopt the proposed approach for molecular dynamics simulations. Second, we plan to account for molecular environments such as a solvent or a protein binding pocket.

ACKNOWLEDGMENTS

The work was supported by a grant for research centers in the field of artificial intelligence, provided by the Analytical Center in accordance with the subsidy agreement (agreement identifier 000000D730321P5Q0002) and the agreement with the Ivannikov Institute for System Programming of dated November 2, 2021 No. 70-2021-00142.

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

## A  EXPERIMENTAL SETUP

Our implementation of GOLF is based on `Schnetpack2.0` (Schütt et al., 2023). Namely, we use `Schnetpack2.0`'s implementation of PaiNN and the data processing pipeline. All the experiments were carried out on a cluster with 2 Nvidia Tesla V100 and 960 Intel(R) Xeon(R) Gold 2.60Hz CPU-cores, and the total computational cost is $\approx 80$ CPU-years and $\approx 1900$ GPU-hours.

To train $f^{\text{GOLF-*}}(\cdot; \boldsymbol{\theta})$, we use a batched version of Algorithm 1 that simultaneously generates several NNP-optimization trajectories with the same NNP and calculates energies and forces using "number of parallel $\mathcal{O}_G$" DFT-workers running in parallel. We use a smaller value of "number of parallel $\mathcal{O}_G$"$= 48$ for $f^{\text{GOLF-1k}}(\cdot; \boldsymbol{\theta})$ to reduce the number of correlated samples in the replay buffer. To prevent the biasing of the model towards newly collected conformations, we sample 10% of each mini-batch from the initial training dataset $\mathcal{D}_0$ during training.

We list all the hyperparameters in Table 5. When evaluating the NNPs on new molecules, we do not employ the $\mathcal{O}_S$ to terminate the optimization trajectory and instead use a fixed timelimit $T_{\text{eval}} = 100$.

Table 5: Hyperparameter values for GOLF-10k.

|  | GOLF-10k |
| --- | --- |
| **NNP hyperparameters** | |
| Backbone | PaiNN |
| Number of interaction layers | 3 |
| Cutoff radius | 5.0 Å |
| Number of radial basis functions | 50 |
| Hidden size (n_atom_basis) | 128 |
| **Training hyperparameters** | |
| Number of parallel $\mathcal{O}_G$ | 120 |
| Batch size | 64 |
| Optimizer | Adam |
| Learning rate scheduler | CosineAnnealing |
| Initial learning rate | $1 \times 10^{-4}$ |
| Final learning rate | $1 \times 10^{-7}$ |
| Gradient clipping value | 1.0 |
| Weight coefficient $\rho$ | $1 \times 10^{-2}$ |
| Total number of training steps | $5 \times 10^5$ |
| Number of additional GO interactions $K$ | 10000 |
| Update-to-data ratio $U$ | 50 |
| Timelimit $T_{\text{train}}$ | 100 |
| Timelimit $T_{\text{eval}}$ | 100 |
| **Conformation optimizer hyperparameters** | |
| Conformation optimizer | L-BFGS |
| Optimization rate $\alpha$ | 1.0 |
| Max number of iterations in the inner cycle | 5 |

## B  EXTERNAL OPTIMIZERS

The external optimizer **Opt** is a crucial component of GOLF, as it generates the NNP-optimization trajectories from which we sample the additional training data. To test the effect of the external optimizer on the training and the evaluation of NNPs, we conduct a series of experiments with SGD (Robbins & Monro, 1951) with momentum, Adam (Kingma & Ba, 2014), and L-BFGS (Liu & Nocedal, 1989). We use the same optimizer for the

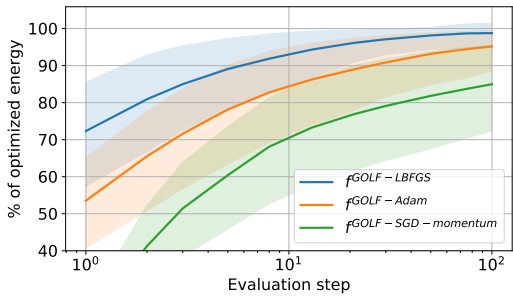 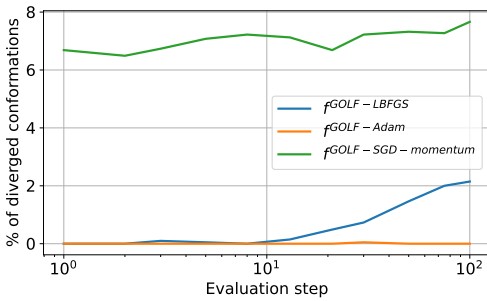

Figure 3: $\overline{\text{pct}}_t$ and $\text{pct}_{\text{div}}^t$, $t = 1, 2, 3, 5, 8, 13, 21, 30, 50, 75, 100$. Shaded regions indicate the 10th and the 90th percentiles of the $\text{pct}(s_t)$, $s \in \mathcal{D}_{\text{test}}$ distribution. The x-axis is log-scaled.

Table 6: Hyperparameter values for GOLF with different external optimizers.

|  | GOLF-LBFGS | GOLF-Adam | GOLF-SGD-momentum |
|---|---|---|---|
| **Training hyperparameters** |  |  |  |
| Total number of training steps | $2 \times 10^5$ | $2 \times 10^5$ | $2 \times 10^5$ |
| Update-to-data ratio $U$ | 20 | 20 | 20 |
| Timelimit $T$ (training) | 100 | 200 | 200 |
| Timelimit $T$ (evaluation) | 100 | 500 | 500 |
| **Conformation optimizer hyperparameters** |  |  |  |
| Conformation optimizer | L-BFGS | Adam | SGD |
| Optimization rate $\alpha$ | 1.0 | $5 \times 10^{-3}$ | $5 \times 10^{-3}$ |
| Max number of iterations in the inner cycle | 5 | – | – |
| Momentum | – | – | 0.9 |

training and the evaluation of NNPs. We dub the resulting models as $f^{\text{GOLF-10k-SGD}}(\cdot; \boldsymbol{\theta})$, $f^{\text{GOLF-10k-Adam}}(\cdot; \boldsymbol{\theta})$ and $f^{\text{GOLF-10k-LBFGS}}(\cdot; \boldsymbol{\theta})$ respectively. As the `pytorch` implementation of L-BFGS includes an inner cycle with up to 5 (empirically chosen hyperparameter) NNP evaluations, we run $f^{\text{GOLF-10k-Adam}}(\cdot; \boldsymbol{\theta})$ and $f^{\text{GOLF-10k-SGD}}(\cdot; \boldsymbol{\theta})$ for 500 steps instead of 100 for $f^{\text{GOLF-10k-LBFGS}}(\cdot; \boldsymbol{\theta})$. We train such models for $2 \times 10^5$ training steps instead of $5 \times 10^5$ to save computational resources. We provide training hyperparameters for $f^{\text{GOLF-10k-*}}(\cdot; \boldsymbol{\theta})$ with different external optimizers in Table 6 and omit hyperparameters identical to those in Table 5. Such a number of training steps is enough to show the superiority of the L-BFGS external optimizer compared to other optimizers.

As it can be seen in Figure 3, $f^{\text{GOLF-10k-LBFGS}}(\cdot; \boldsymbol{\theta})$ outperforms other optimizers in terms of $\overline{\text{pct}}_T$. However, $f^{\text{GOLF-10k-Adam}}(\cdot; \boldsymbol{\theta})$ performs better in terms of $\text{pct}_{\text{div}}$. We hypothesize that $f^{\text{GOLF-10k-Adam}}(\cdot; \boldsymbol{\theta})$ can be tuned to match the optimization quality of $f^{\text{GOLF-10k-LBFGS}}(\cdot; \boldsymbol{\theta})$, while retaining close-to-zero $\text{pct}_{\text{div}}$, but leave this for future work.

## C MSE FOR GOLF

To further show that $f^{\text{GOLF-10k}}(\cdot; \boldsymbol{\theta})$ and $f^{\text{traj-500k}}(\cdot; \boldsymbol{\theta})$ perform similarly, we evaluate the prediction quality of $f^{\text{GOLF-10k}}(\cdot; \boldsymbol{\theta})$ along the NNP-generated trajectories and plot the MSE for predicted energies and forces in Figure 4.

## D NABLADFT DATASET

Throughout this work, we use several subsets of nablaDFT (Khrabrov et al., 2022) dataset. The nablaDFT dataset is based on the Molecular Sets (MOSES) dataset, which is a diverse subset of the ZINC dataset, con-

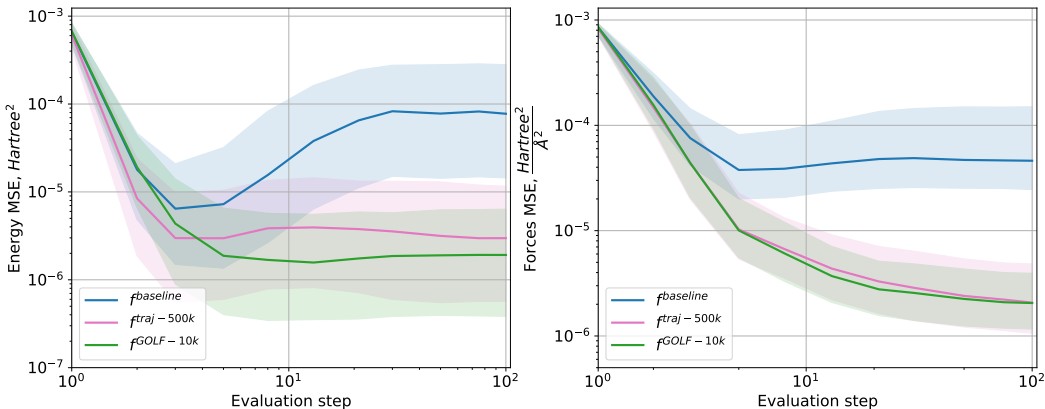

Figure 4: Mean squared error (MSE) of energy and forces prediction for NNPs trained on $\mathcal{D}_0$ and $\mathcal{D}_{\text{traj-500k}}$, and NNP trained with GOLF. To compute the MSE, we collect NNP-optimization trajectories of length $T = 100$ and calculate the ground truth energies and forces in steps $t = 1, 2, 3, 5, 8, 13, 21, 30, 50, 75, 100$. Solid lines indicate the median MSE, and the shaded regions indicate the 10th and the 90th percentiles. Both the x-axis and y-axis are log scaled

taining approximately one million drug-like molecules with atoms C, N, S, O, F, Cl, Br, and H. For each molecule from the dataset, the authors ran the conformation generation method Wang et al. (2020) from the `RDKit` software Landrum et al. (2022). Next, they clustered the resulting conformations with the Butina clustering method Barnard & Downs (1992). Lastly, they selected the smallest number of clusters that cover at least 95% of conformations and used their centroids as a set of conformations for a given molecule. This procedure has resulted in 1 to 62 unique conformations for each molecule, with 5 340 152 total conformations in the full dataset. Finally, these conformations were evaluated with a DFT-based oracle. The baselines and GOLF models are trained on the train set $\mathcal{D}_0$: a subset of nablaDFT, which contains 4 000 molecules and $\approx 10\,000$ conformations ( 2.5 conformations per molecule). The test set $\mathcal{D}_{\text{test}}$ contains $\approx 10\,000$ different molecules and 19 447 conformations. Optimization trajectories for $f^{\text{traj-}}(\cdot; \boldsymbol{\theta})$ were obtained with a DFT-based oracle by optimizing conformations from the train set. The average length of a trajectory is $\approx 100$ steps. Finally, generative baselines were trained to map conformations from $\mathcal{D}_0$ to their optimal counterparts.

## E   SPICE DATASET

Another dataset that is used in our work is SPICE (Eastman et al., 2023). It is a subset of the PubChem dataset (Kim et al., 2023) and contains a diverse set of drug-like molecules. The total number of molecules in SPICE is 14644. The dataset contains 25 high-energy conformations and 25 low-energy near-optimal conformations per molecule. Molecules contain the following atoms: C, N, S, O, F, Cl, Br, I, P, and H. To cross-validate models trained on SPICE and nablaDFT, we filtered out molecules containing I and P atoms. This procedure resulted in 13 231 filtered molecules. To make the training setup consistent with nablaDFT, we selected $\approx 3500$ molecules and $\approx 9500$ conformations for the SPICE training set $\mathcal{D}_0^{\text{SPICE}}$. The training set only contains the high-energy conformations, as we observed that training on near-optimal conformations leads to instabilities. The test set $\mathcal{D}_{\text{test}}^{\text{SPICE}}$ includes $\approx 7000$ molecules and $\approx 18000$ conformations. The test set contains both high-energy and low-energy conformations in equal parts. Note that initially, $\mathcal{D}_{\text{test}}^{\text{SPICE}}$ was supposed to match the size of $\mathcal{D}_{\text{test}}$ but the DFT-based optimization for some molecules did not converge, so we excluded them from the test set. Similar to Section D, optimization trajectories were obtained with a DFT-based oracle by optimizing conformations from $\mathcal{D}_0^{\text{SPICE}}$. The only difference is that we used optimization in spherical coordinates instead of Cartesian. The change of coordinates resulted in shorter optimization trajectories (around 25 steps on average). The biggest trajectories dataset for SPICE $\mathcal{D}_{\text{traj-220k}}^{\text{SPICE}}$ thus only contains $\approx 220\,000$ conformations.

## F   DISTRIBUTION MATCHING METRICS

Consider the evaluation of the NNP on the dataset $\mathcal{D}_{\text{test}}$. Let $\mathbb{S}_g = \{s_T\}_{s \in \mathcal{D}_{\text{test}}}$ denote the set of all final conformations in the NNP-optimization trajectories, and $\mathbb{S}_r = \{s_{\mathbf{Opt}}^{\text{DFT}}\}_{s \in \mathcal{D}_{\text{test}}}$ denote the set of all ground truth optimal conformations obtained by the GO. To measure the difference between $s \in \mathbb{S}_g$ and $\tilde{s} \in \mathbb{S}_r$, we use the `GetBestRMSD` in the RDKit package and denote the root-mean-square deviation as $\text{RMSD}(s, \tilde{s})$. The

recall-based coverage and matching scores are defined as follows:

$$\mathrm{COV}(\mathbb{S}_g, \mathbb{S}_r) = \frac{1}{|\mathbb{S}_r|} \left| \{ s \in \mathbb{S}_r \mid \mathrm{RMSD}(s, \tilde{s}) < \delta, \exists \tilde{s} \in \mathbb{S}_g \} \right|;$$

$$\mathrm{MAT}(\mathbb{S}_g, \mathbb{S}_r) = \frac{1}{|\mathbb{S}_r|} \sum_{s \in \mathbb{S}_r} \min_{\tilde{s} \in \mathbb{S}_g} \mathrm{RMSD}(s, \tilde{s}).$$

(9)

COV is number of conformations in $\mathbb{S}_r$ that are "reasonably" close ($\mathrm{RMSD} < \delta$) to some conformation from $\mathbb{S}_s$. MAT is the average over all $s \in \mathbb{S}_r$ RMSD to the closest conformation from $\mathbb{S}_g$. Note that both COV and MAT are not ideal metrics for the optimization task because they do not consider the energy of the final conformation.

## G  GENERATIVE BASELINES

Table 7: Energy and Recall-based scores. We set $\delta = 0.5$Å when computing the COV.

| Methods | $\overline{\mathrm{pct}}_T(\%)\uparrow$ | $\mathrm{pct}_{\mathrm{div}}(\%)\downarrow$ | COV(%)↑ Mean | MAT (Å)↓ Mean |
|---|---|---|---|---|
| TD | $24.04 \pm 21.3$ | 54.1 | 12.53 | 1.284 |
| ConfOpt | $33.36 \pm 22.0$ | 92.5 | 24.08 | 1.004 |
| Uni-Mol+ | -* | -* | 13.49 | 1.25 |
| $\mathrm{TD}_{pr}$ | $25.63 \pm 21.4$ | 46.9 | 11.25 | 1.33 |
| $\mathrm{ConfOpt}_{pr}$ | $36.48 \pm 23.0$ | 84.5 | 19.88 | 1.05 |
| Uni-Mol+$_{pr}$ | $69.9 \pm 23.1$ | 23.2 | 15.29 | 1.23 |
| Uni-Mol+$_{init}$ | $54.92 \pm 20.5$ | 8.0 | 63.41 | 0.44 |
| Uni-Mol+$_{pr+init}$ | $62.20 \pm 17.2$ | 2.8 | 68.79 | 0.407 |

* The energy-based metrics for the Uni-Mol+ model are not reported due to the problems with energy computation.

In this section, we provide additional details considering the training of generative baselines and the corresponding metrics (see Table 7). We consider three architectures designed for conformation generation (Energy-inspired molecular conformational optimization (ConfOpt) (Guan et al., 2021), Torsional diffusion (TD) (Jing et al., 2022), and Uni-Mol+ (Lu et al., 2023)) and adapt them to the task of geometry optimization. For the first two models, we follow the same setup proposed in the corresponding papers and train models to generate optimal conformations from the ones generated by RDKit. In the case of Uni-Mol+, we compare two setups: i) the model is trained to generate optimal conformations conditioned on geometries from RDKit; ii) the model is trained to generate optimal conformations conditioned on non-optimal conformations from nablaDFT. We add a subscript $init$ in the latter case. Moreover, we also experiment with starting the training with randomly initialized weights and pretrained checkpoints. We use a checkpoint obtained on the PCQM4MV2 dataset (Nakata & Maeda, 2023) in the case of Uni-Mol+ and on the GEOM-DRUGS dataset (Axelrod & Gomez-Bombarelli, 2022) otherwise. We add a subscript $pr$ for pre-trained models.

To save computational resources, all the models from Table 7 were evaluated on a subset of $\mathcal{D}_{\mathrm{test}}$ that we call $\mathcal{D}_{\mathrm{test}}^{\mathrm{small}}$ ($|\mathcal{D}_{\mathrm{test}}^{\mathrm{small}}| = 2044$). Our findings are as follows: ConfOpt and TD models perform much worse on the energy optimization task in our setup than on the tasks reported in the corresponding papers. Uni-Mol+ performs on par with NNP baselines but worse than the models trained on additional data. We suspect the reason for such behavior of TD is the small amount of data and the necessity to model a discrete distribution over the optimal geometries instead of the whole conformational space. The TD authors also report that the resulting conformations differ by a large margin in terms of energies and other quantum chemical properties from the reference conformations and require additional optimization with the simulator. We hypothesize that in the case of ConfOpt, the main problems are the choice of architecture and the fact that the model generates optimal conformations from SMILES and does not use initial geometries.

In Table 7, we observe that i) all generative baselines benefit in terms of $\mathrm{pct}_{\mathrm{div}}$ from using pre-trained weights. Even though the pre-training was done using data generated by DFT-based methods with different levels of theory than in nablaDFT; ii) starting from non-optimal conformations from nablaDFT greatly benefits all metrics for Uni-Mol+, indicating that a reasonable initial conformation is crucial for generative baselines.

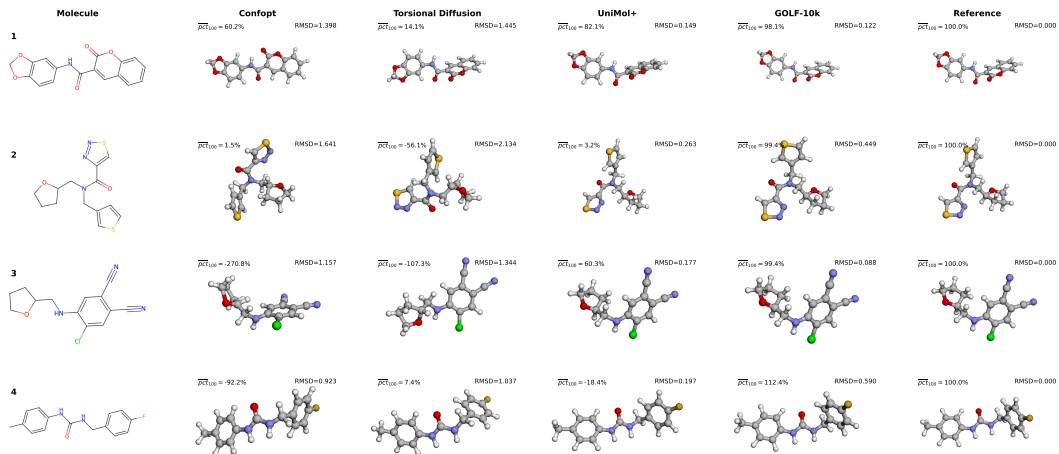

Figure 5: Visualization of final conformations obtained by various models, the 2D view of the molecule, and the reference optimal conformation obtained with the $\mathcal{O}_G$ .

## H    FINAL CONFORMATIONS COMPARISON

To highlight the difference in the quality of conformation optimization, we visualize final conformations for ConfOpt, Torsional Diffusion, Uni-Mol+, and our best-performing model (GOLF-10k) with py3Dmol (Rego & Koes, 2015). In Figure 5, we provide visualizations for 4 molecules from the test set $\mathcal{D}_{\text{test}}$. We also provide the 2D visualization of the molecule obtained with RDKit and a visualization of the reference optimal conformation obtained with the $\mathcal{O}_G$ .

Molecules 1 and 3 are an example of the case where the conformation optimization with GOLF-10k converges to the same local minima as $\mathcal{O}_G$ : the RMSD to the reference conformation is close to zero, while the $\overline{\text{pct}}_{100}$ is close to 100%. It is also hard to spot any visual differences. On the other hand, molecules 2 and 4 illustrate the case where the conformation optimization with GOLF-10k converges to the different local minima: RMSD is larger than zero, but the $\overline{\text{pct}}_{100}$ is 100% or even greater than 100% in case of the molecule 4. The visual difference between the resulting conformations is prominent.

Negative values for the $\overline{\text{pct}}_{100}$ are often caused by distorted distances between atoms in cycles (ConfOpt optimization for molecules 3 and 4). Low positive values of $\overline{\text{pct}}_{100}$ generally indicate conformations with correct distances between atoms but incorrect dihedral angles between different parts of the molecule (ConfOpt optimization for molecule 2, Torsional diffusion for molecule 1, Uni-Mol+ optimization for molecule 2).

