# OpenReview forum: "Gradual Optimization Learning for Conformational Energy Minimization"
_ICLR.cc/2024/Conference — ICLR 2024 poster_

### Official Review · Reviewer_RbRE · 2023-11-01

**Soundness:** 2 fair
**Presentation:** 4 excellent
**Contribution:** 3 good
**Rating:** 8
**Confidence:** 4

**Summary:**

The authors propose an active learning approach to train neural network potentials by employing a cheap surrogate oracle before querying the much more expensive genuine oracle.

**Strengths:**

The paper is very well written with clear insight and motivation. The method is very neat and the results are promising. I am excited to see more work that follows these mixed genuine and surrogate oracle approach.

**Weaknesses:**

Additional baselines and metrics could make the result stronger.

In terms of the baseline, the true innovation of the framework is the "active learning" component, and not "conformer generation" itself. Therefore, the baseline conversation probably should focus on other active learning approaches such as Kulichenko et al (2023) or Chem et al. (2019). While these methods require OG, the author can still compare to these methods by contrasting the OG query budget to the same amount, but use random selection instead of SG estimates as proposed by GOLF. The comparison to TD/ConfOpt, while interesting, the problem setup and training data requirement are very different from what the authors are trying to demonstrate here.

In terms of the metrics, it would be very helpful if the authors can provide more context about why "percentage of minimized energy" is a meaningful metric, and why >98% is considered solving the optimization. If >98% is broadly considered as solving the optimization, can the authors report what percentage of targets in the test set is "solved" under different experiment setup?

**Questions:**

I would consider raising my score if the authors can address my concerns around baselines and metrics as mentioned in the Weaknesses section.

---

> ### Author Response · Authors · 2023-11-23
> **Answer to Official Review of Submission9224 by Reviewer RbRE**
>
> - **Weakness 1**
>
> > In terms of the baseline, the true innovation of the framework is the "active learning" component, and not "conformer generation" itself. Therefore, the baseline conversation probably should focus on other active learning approaches such as Kulichenko et al (2023) or Chem et al. (2019). While these methods require OG, the author can still compare to these methods by contrasting the OG query budget to the same amount, but use random selection instead of SG estimates as proposed by GOLF.
>
> The method (UDD) presented in Kulichenko et al. (2023) is designed for the exploration of the configuration space of a **single** molecule. The method generates new data by running molecular dynamics from a set of initial conformations. To better explore the space, the energy in molecular dynamics is augmented with a $E_{\text{bias}}$ term that encourages the MD to sample conformations from regions of configuration space where the ensemble of NNPs is uncertain. In theory, this approach may also improve the atomic forces prediction, which would help the solution of the optimization task, but it remains to be studied. However, the UDD requires training a new NNP ensemble for each new molecule, which is a completely different problem setup compared with the one in our study. While it is an interesting task to adjust the UDD for the multi-molecule setup, we believe it is out of the scope of the current study.
>
> If we understood you correctly, by Chem et al.(2019) you meant Chan et al. (2019) - "Bayesian optimization for conformer generation". While interesting, this approach requires DFT calculations during the inference, whereas GOLF only interacts with the oracle during the training phase and thus is more efficient.
>
> - **Weakness 2**
>
> > The comparison to TD/ConfOpt, while interesting, the problem setup and training data requirement are very different from what the authors are trying to demonstrate here.
>
> We agree that the generative setting is sufficiently different from the iterative optimization one. However, it is possible to formulate the problem of finding optimal geometries as a generative task. If it is possible to generate optimal geometries directly (e.g., without further relaxation), our framework would be superfluous. Contrary to this, our experiments show that it is still unclear if the generative models can achieve comparable performance.
>
> - **Weakness 3**
>
> > In terms of the metrics, it would be very helpful if the authors can provide more context about why "percentage of minimized energy" is a meaningful metric, and why >98% is considered solving the optimization. If >98% is broadly considered as solving the optimization, can the authors report what percentage of targets in the test set is "solved" under different experiment setup?
>
> Thank you very much for pointing this out! Generally accepted chemical precision is 1 kcal/mol [1]. The average total optimized energy is 43.2 kcal/mol therefore 2% is about the chemical precision. We removed the "2%" from the manuscript as it is indeed more correct to reason about being “*on par*” with the optimizer in terms of chemical precision. Following your suggestion, we decided to add the percentage of “solved” conformations (we call it $\operatorname{pct}\_{\text{solved}}$) as one of the metrics for all the experiments. We found that it helps to better demonstrate the superiority of the GOLF compared to the baseline approaches.
>
> [1] Helgaker, T., Ruden, T. A., Jørgensen, P., Olsen, J., & Klopper, W. (2004). A priori calculation of molecular properties to chemical accuracy. *Journal of Physical Organic Chemistry*, *17*(11), 913-933.

---

### Official Review · Reviewer_HQch · 2023-11-01

**Soundness:** 3 good
**Presentation:** 3 good
**Contribution:** 3 good
**Rating:** 6
**Confidence:** 3

**Summary:**

In the paper, the authors present Gradual Optimization Learning Framework (GOLF), a framework for improving the efficiency of generating low-energy molecular conformation prediction models, a crucial technology used in computer-aided drug discovery and materials design.

Overall, the paper is well written and presents some valuable insights into applying active learning efficiently for the discovery of energy minimized conformations. Traditional approaches, such as Density-functional theory (DFT) models, use high-fidelity physics-based numerical quantum chemistry simulators whose computational costs are exponential with respect to the complexity of the molecule under study. Unfortunately, this limits their applicability to simple molecules with few atoms or electrons. To address computational complexity and to scale conformal optimization to more complex molecules, researchers have explored various alternatives based on lower fidelity linear models and, more recently, neural network models that leverage the availability of computed quantum property molecular databases. Sadly, these alternate approaches lead to inaccurate predictions and suffer from distribution shift. To scale conformal energy prediction to larger molecules while addressing computational cost, the authors propose GOLF, an automated data augmentation scheme and a hybrid computational approach that combines the use of both high and low-fidelity simulators as well as neural networks.

In Section 1, the authors present the concept of low-energy molecular conformations and their uses. This section is well written and provides an adequate background of the approach and the insights that motivate the solution. The authors indicate that the fundamental problem with traditional approaches, such as DFT, to obtain optimal conformations is their high computational cost. These approaches, which are based on numerical quantum chemistry simulations that calculate anti-gradients representing molecular forces, are iterative by nature and, when given a sufficiently complex molecule or physical system, may fail to complete even a single iteration. For this reason, the authors claim that “reducing the number of interactions with the physical simulator” is crucial for efficiency. The authors then go on to describe current methods that apply Neural Network Potentials (NNP), a form of deep neural networks, to the problem. NNP-based techniques significantly reduce computational complexity by using the gradients inherent to neural networks to model the molecular forces, thereby obviating the need for expensive simulations. Unfortunately, the NNP approach suffers from distribution shift resulting in inaccurate predictions. The authors then introduce GOLF, which employs a data augmentation active learning scheme to improve the diversity of the training dataset, thereby alleviating the distribution shift. By doing so, GOLF achieves energy minimized conformation prediction accuracy comparable with that of high-fidelity simulations while retaining the intrinsic efficiencies gained by using neural networks. A novel framework for data-efficient training of NNPs, GOLF comprises three components: 1) a computationally expensive high-fidelity simulation that is a genuine oracle (GO) used to calculate the ground truth energies and forces; 2) an optimizer that uses the NNP gradients to produce optimization trajectories that constitute the additional training dataset; and 3) a computationally inexpensive low-fidelity simulation that is a surrogate oracle (SO) used to augment the training dataset. Finally, the authors then conclude this section by summarizing their contributions.

However, there are various weaknesses in this section as well that can be addressed to improve the quality of the paper. 1) The statement that “reducing the number of interactions with the physical simulator” is unclear and the reviewer assumes the efficiency objective is attained by reducing the number of iterations required to produce optimal low-energy conformations. 2) The authors state that they augment the dataset with “optimization trajectories” without explaining what such a trajectory is and how the trajectories address distribution shift. At a minimum, providing a reference to the discussion of augmented data in Section 5 would be useful. 3) Moreover, as GOLF requires running the high-fidelity simulation, GO, to produce the anti-gradients, it is unclear whether GOLF can be successful when GO fails to complete a single iteration. This is a serious flaw in the paper and needs to be addressed. 4) In addition, the “Ab initio property” phrase is used without a definition or description and seems rather superfluous to the narrative. 5) The adequacy of the requirement for 5 X 105  “additional oracle interactions”, which presumably means optimization trajectories that augment the training dataset, is likely anecdotal based on the molecules selected for the experiments. If that is not the case, an explanation of why it is generally applicable should be articulated.

In the “Related Works” section (Section 2), the authors describe a variety of contemporary approaches to conformation generation. The benefits and drawbacks of these methods are discussed. However, it is not clear to the reviewer how GOLF addresses the drawbacks of these approaches. Successfully addressing the drawbacks could demonstrate GOLF’s superiority. Also, form the exposition in this section, it is not clear how significantly different the GOLF approach is to the active learning technique presented by Kulichecnko et al. (2023). Finally, the phrase “we believe it is necessary to explore further the ability …” is confusing. Are the authors proposing future work or teeing up the discussion in the remainder of the paper?

In the “Notations and Preliminaries” section (Section 3), the authors summarize the theoretical foundation of their approach. Although informative, the notation is somewhat cryptic and can benefit from slightly greater verbosity or additional graphics. Also, mentioning GOLF models in this section, without any discussion as to what they are or how they differ from ftraj, seems premature and confusing. At the very least, there should be a forward reference to Section 5 that articulates how GOLF intelligently identifies the datasets that promote diversity, which enhances prediction performance. Moreover, a small discussion of the NNP architecture used in the experimentation would be useful for the sake of completeness.

Section 4 presents “Conformation Optimization”. This section is well written and the both the graphic, Figure 1, and the table, Table 1, provide valuable insight. The Figure 1 graphic clearly depicts the distribution shift, in terns of Mean Square Error (MSE), increasing as the optimization progresses. The graphic also depicts that the prediction accuracy improves – MSE decreases – when augmenting the training dataset with GO produced optimization trajectories. This is an important result but without highlighting it, the reader can easily miss that it is one of the contributions of the paper. Table 1 seems to be highlighting precision of the approach, but the word is not used in the discussion. It is unclear to the reviewer as to the innovativeness of the approach, which may be construed as a weakness. Moreover, using the GO to provide the baseline training dataset may limit the scalability of the approach.

The authors present a sound argument in Section 5 where they present the GOLF algorithm and discuss using a high-performance low-cost surrogate oracle to make the data generation computationally tractable. The “Experiments” section, Section 6 is reasonably complete though much of the discussion seems anecdotal. The reviewer is unable to determine how many of the experimental results were achieved through a fortuitous selection of the molecules under study. Also, the authors report that the GOLF technique can produce “a high percentage of diverged conformations”. This is not surprising as the training dataset is likely to be much noisier as a result of the choice to use a low-cost simulation. It would be nice to get better characterization of the noise and its effects, including the loss in efficiency resulting from these unusable conformations.

Sections 7, 8, and Appendices conclude the paper. An explicit tie back to the goals and contributions identified in the Introduction would be beneficial.

**Strengths:**

Overall, the paper is well written and informative. It seems relevant to improving the computational tractability of conformational energy minimization. The insight to use active learning to address the distribution shift and improve accuracy is valuable. Also, the approach to active learning by using low-cost simulation to augment the training data set without impacting the quality of the subsequent model is somewhat innovative.

**Weaknesses:**

The various weaknesses are already detailed above. Here I summarize the most important ones. Using GO to generate the baseline dataset may limit the scalability of the approach. It is unclear why the authors do not use the molecular databases they mention in the “Introduction” section to extract the baseline dataset. Some of the discussion is somewhat cryptic and can benefit from some additional discussion or graphics. The results seem anecdotal, tied to the selected dataset and molecules, and thus may not generalize particularly well.

**Questions:**

Suggestions:
1) Clean up the use of "interactions with the physical simulator" and the "number of iterations" in several locations in the paper. They seem to imply the same concept. If they are, just use a single phrase for both.
2) Additional references early on in the paper to the results later on in the paper will help the reader question many of the seemingly unsupported statements.
3) A better tie-in to how GOLF addresses the drawbacks of the "Related Works" section will improve the quality of  the paper.
4) Reduce the amount of mathematical notation in the “Notations and Preliminaries” section (Section 3) to simplify the narrative and improve understandability.
5) The main result in Section 4, the decrease in MSE by augmenting the dataset, should be highlighted and tied-in to the wording of the contributions listed in the introduction section of the paper.

Questions:
1) GOLF requires running the high-fidelity simulation, GO, to produce the anti-gradients initial training data. In a different section of the paper, the claim is that for a sufficiently complex system, the physical simulation may not succeed in completing even a single iteration in a reasonable amount of time. Taken together, these two statements seem to suggest that GOLF is atomic complexity scale limited. How do the authors claim to address this apparent limitation?
2) In the Experiments section (Section 6), how much of the results are related to the selection of the molecules under study? Stated differently, how do the authors plan to address the generalizability of the approach?
3) The additional data generated for Active Learning are selected based only on errors. Without some sort of approach to balance the introduction of new data, does the approach bias the dataset distribution causing the learned distribution to fail to generalize to other molecules?
4) How different is the GOLF approach from the active learning technique presented by Kulichecnko et al. (2023).

---

> ### Author Response · Authors · 2023-11-23
> **Answer to Official Review of Submission9224 by Reviewer HQch (1/4)**
>
> We appreciate your suggestions on improving the writing of the introduction. We found them very useful and incorporated the proposed modifications in the revised manuscript. We summarize the changes below:
>
> ## Introduction
>
> > The statement that “reducing the number of interactions with the physical simulator” is unclear
> >
>
> We replaced this sentence with the following: “Therefore, it is crucial to develope alternative approaches (such as Neural Networks-based) that reduce the computational complexity of iterative optimization.”
>
> > The authors state that they augment the dataset with “optimization trajectories” without explaining what such a trajectory is and how the trajectories address distribution shift.
> >
>
> We added a reference to Section 4.
>
> > Moreover, as GOLF requires running the high-fidelity simulation, GO, to produce the anti-gradients, it is unclear whether GOLF can be successful when GO fails to complete a single iteration.
> >
>
> We reworked this sentence: “However, for large molecules even a single iteration may take up several hours of CPU-compute.“
>
> Considering your question, refer to the answer to Question 1.
>
> > In addition, the “Ab initio property” phrase is used without a definition or description and seems rather superfluous to the narrative.
> >
>
> “*Ab initio*” is a latin term that means “from first principles”. It is widely used in quantum chemistry literature implying that the only inputs into an *ab initio* calculation are physical constants.
>
> > The adequacy of the requirement for $5 \times 10^5$ “additional oracle interactions”, which presumably means optimization trajectories that augment the training dataset, is likely anecdotal based on the molecules selected for the experiments. If that is not the case, an explanation of why it is generally applicable should be articulated.
>
> We agree that this constant may not hold on another dataset. However, we believe that the reduction in the amount of required oracle interactions will still be significant. We decided to remove $5 \times 10^5$ from the abstract and the intro, replacing it with a more appropriate statement.
>
> ## Notation and preliminaries
>
> > The notation is somewhat cryptic and can benefit from slightly greater verbosity or additional graphics.
>
> We reworked the notation in the revised version of the manuscript.
>
> > Also, mentioning GOLF models in this section, without any discussion as to what they are or how they differ from ftraj, seems premature and confusing.
>
> We added a reference to Section 5.
>
> > Moreover, a small discussion of the NNP architecture used in the experimentation would be useful for the sake of completeness.
>
> We listed all the hyperparameters in Appendix.
>
> ## Conformation Optimization with NNPs
>
> > This is an important result but without highlighting it, the reader can easily miss that it is one of the contributions of the paper.
>
> We referenced Figure 1 in the contributions section.
>
> > It is unclear to the reviewer as to the innovativeness of the approach, which may be construed as a weakness. Moreover, using the GO to provide the baseline training dataset may limit the scalability of the approach.
>
> While working on the topic of molecular optimization, we found that this area is severely under-researched. To the best of our knowledge, there are no papers that present the idea of enriching the training dataset with optimization trajectories. However, in this year Open Catalyst challenge [https://opencatalystproject.org/challenge.html], a similar dataset of optimization trajectories for adsorbate-catalyst pairs has been published. We will cite the OCP23 challenge in the revision.

---

> ### Author Response · Authors · 2023-11-23
> **Answer to Official Review of Submission9224 by Reviewer HQch (2/4)**
>
> ## Experiments
>
> > The “Experiments” section, Section 6 is reasonably complete though much of the discussion seems anecdotal. The reviewer is unable to determine how many of the experimental results were achieved through a fortuitous selection of the molecules under study.
>
> In our experiments, we use a randomly selected subset $\mathcal{D}_0$ of the nablaDFT dataset. The conformations in the nabla DFT dataset are generated based on SMILES representations of molecules from the MOSES dataset. The MOSES dataset consists of a large number of druglike molecules and is considered to be representative.  To ensure reasonable computational cost of DFT computations, the subset  $\mathcal{D}_0$ is limited to molecules with number of atoms < 36. However, apart from the increased computational cost we do not forsee any issues with using GOLF with bigger molecules.
>
> Therefore we significantly extended the main evaluation dataset (10x) by sampling more conformations with less than 35 atoms from the nablaDFT dataset. The extended evaluation dataset contains 20k conformations for X molecules. It contains  $\mathcal{D}\_{\text{test}}$ as its subset.
> Moreover, we selected 1828 molecules (1 conformation per molecule) from the SPICE dataset, optimized them with $\mathcal{O}\_G$ to get the optimal energy. We evaluated all non-generative models from our paper on this new SPICE test set $\mathcal{D}\_{\text{test}}^{\text{SPICE}}$.
> We provide the results in the table below. We understand that the size of the $\mathcal{D}\_{\text{test}}^{\text{SPICE}}$ may raise similar concerns, but we were unable to collect a larger dataset during the rebuttal due to the need for an expensive $\mathcal{O}\_G$ optimizations. We will extend the $\mathcal{D}\_{\text{test}}^{\text{SPICE}}$ to ~20k conformations and provide results for generative models in the camera-ready version.
>
> ### SPICE test (1828 conformations)
> |  | OpenFF | RDKit | $f^{baseline}$ | $f^{traj-10k}$ | $f^{traj-100k}$ | $f^{traj-500k}$ | $f^{GOLF-1k}$ | $f^{GOLF-10k}$ | $SPICE-f^{baseline}$ | $SPICE-f^{GOLF-10k}$ |
> | --- | --- | --- | --- | --- | --- | --- | --- | --- | --- | --- |
> | $\overline{\operatorname{pct}}\_{100}$ | 61.14 | 77.58 | 72.71 | 73.22 | 77.06 | 80.25 | 80.31 | 82.41 | 79.76 | 86.59 |
> | $\operatorname{pct}\_{\text{div}}$ | 5.38 | 8.15 | 27.73 | 17.45 | 16.02 | 12.80 | 13.56 | 12.41 | 23.08 | 15.09 |
> | $\overline{E^{\text{res}}\_{100}}$ | 12.58 | 8.42 | 13.62 | 9.63 | 8.09 | 6.88 | 6.63 | 6.06 | 6.06 | 4.19 |
> | $\operatorname{pct}\_{\text{solved}}$ | 0.05 | 10.72 | 6.35 | 5.17 | 15.90 | 25.65 | 17.53 | 27.73 | 8.91 | 19.41 |
>
> The OpenFF is a non-neural force field used in the original SPICE paper [1]. $SPICE-f^{baseline}$ was trained on ~10000 conformations (2.5 conformations per molecule) from the SPICE dataset. $SPICE-f^{GOLF-10k}$ was trained with the same hyperparameters as $f^{GOLF-10k}$ and used $SPICE-f^{baseline}$ to initialize the NNP. These preliminary results show that models trained on nablaDFT and SPICE datasets demonstrate comparable performance on $\mathcal{D}\_{\text{test}}^{\text{SPICE}}$.
>
> ### Extended test (19477 conformations)
>
> |  | RDKit | $f^{baseline}$ | $f^{rdkit}$ | $f^{traj-10k}$ | $f^{traj-100k}$ | $f^{traj-500k}$ | $f^{GOLF-10k}$ |
> | --- | --- | --- | --- | --- | --- | --- | --- |
> | $\overline{\operatorname{pct}}\_{100}$ | 85.47 | 77.88 | 93.04 | 95.08 | 96.15 | 98.75 | 98.78 |
> | $\operatorname{pct}\_{\text{div}}$ | 0.62 | 7.46 | 4.43 | 4.50 | 2.77 | 1.99 | 2.98 |
> | $\overline{E^{\text{res}}\_{100}}$ | 5.50 | 8.58 | 2.77 | 1.96 | 1.50 | 0.53 | 0.51 |
> |  $\operatorname{pct}\_{\text{solved}}$ | 4.05 | 8.18 | 35.42 | 37.01 | 52.71 | 73.41 | 77.26 |
>
> These results show that all conclusions made in the first version of the paper hold on an extended evaluation dataset.
>
> [1] Eastman, P., Behara, P. K., Dotson, D. L., Galvelis, R., Herr, J. E., Horton, J. T., ... & Markland, T. E. (2023). Spice, a dataset of drug-like molecules and peptides for training machine learning potentials. Scientific Data, 10(1), 11.

---

> ### Author Response · Authors · 2023-11-23
> **Answer to Official Review of Submission9224 by Reviewer HQch (3/4)**
>
> > Also, the authors report that the GOLF technique can produce “a high percentage of diverged conformations”. This is not surprising as the training dataset is likely to be much noisier as a result of the choice to use a low-cost simulation. It would be nice to get better characterization of the noise and its effects, including the loss in efficiency resulting from these unusable conformations.
> >
>
> First, we would like to stress that we do not use the low-cost simulation neither to estimate energies/forces for the newly collected data, nor to generate conformations for training. To estimate energies and forces, we use DFT. Additional conformations for training are generated during the optimization with the NNP. The low-cost simuator is used to select a single conformation from the optimization trajectory. This selected conformation is then evaluated with the DFT-based oracle and added to the training set.
>
> Second, the “high percentage of diverged conformations” refers to the experiment in Section 6.4 (applying NNP trained with GOLF  to larger molecules than in the training dataset). In our code, the optimization is considered diverged either when the initial energy is lower than the final energy or when the DFT-worker is timed out. For small molecules in $\mathcal{D}_0, \mathcal{D}_{\text{test}}$ this happens when the resulting conformation is poor, but we found that for larger molecules this can also happen with reasonable near-optimal conformations. We also found that increasing the timeout value solves this problem (see updated table in section 6.4).
>
> We extended the $\mathcal{D}_{\text{LM}}$ to 2000 molecules (1 conformation per molecule) and re-evaluated NNPs with the increased timeout value. We updated section 6.4 in the revision. As it can be seen from the updated Table 3, $f^{\text{GOLF-10k}}$ on large molecules experiences a small performance drop, while retaining the same percentage of diverged optimizations.
>
> ## Weaknesses
>
> - **Weakness 1**
>
> > Using GO to generate the baseline dataset may limit the scalability of the approach. It is unclear why the authors do not use the molecular databases they mention in the “Introduction” section to extract the baseline dataset.
>
> First, the limited scalability of this approach is stated as the main motivation for our proposed GOLF framework. Second, we would like to note that we are not aware of any molecular databases that contain optimization trajectories. In our paper, we identify enriching the training dataset $\mathcal{D}_0$ with optimization trajectories as a crucial step to overcome the distribution shift. In Figure 1, we illustrate that enriching the training dataset with optimization trajectories helps alliviate the distribution shift. We agree, that in theory, datasets containing both near-optimal and sub-optimal conformations can be used to improve the optimization quality of NNPs. To test that, we trained a baseline model $\operatorname{SPICE}-f^{\text{baseline}}$ on a subset of SPICE dataset and showed that it suffers from the similar issues (see answer to “Experiments” section).
>
> - **Weakness 2**
>
> > Some of the discussion is somewhat cryptic and can benefit from some additional discussion or graphics.
>
> We appreciate your suggestions on improving the writing of our paper! Following various ideas proposed by the reviewers we have reworked the manuscript and uploaded a corrected version as a revision. If you still have any concerns or suggestions, please let us know. We will be happy to improve the writing further.
>
> - **Weakness 3**
>
> > The results seem anecdotal, tied to the selected dataset and molecules, and thus may not generalize particularly well
>
> In order to resolve this concern, we decided to provide additional experiments both on a bigger subset of nablaDFT dataset and a subset of SPICE dataset. See answer to “Experiments” section for more details.

---

> > ### Author Response · Authors · 2023-11-23
> > **Answer to Official Review of Submission9224 by Reviewer HQch (4/4)**
> >
> > ## Questions
> >
> > - **Question 1**
> >
> > > GOLF requires running the high-fidelity simulation, GO, to produce the anti-gradients initial training data. In a different section of the paper, the claim is that for a sufficiently complex system, the physical simulation may not succeed in completing even a single iteration in a reasonable amount of time. Taken together, these two statements seem to suggest that GOLF is atomic complexity scale limited. How do the authors claim to address this apparent limitation?
> >
> > Yes, the GOLF indeed requires a high-fidelity simulator to collect the training data which may may fail on a sufficiently complex system. However, any deep learning model that aims to match the quality of the physical simulator is subject to the same limitation. Motivated by this limitation, we tested the generalization of NNPs trained with GOLF to larger molecules (see section 6.4).
> >
> > - **Question 2**
> >
> > > In the Experiments section (Section 6), how much of the results are related to the selection of the molecules under study? Stated differently, how do the authors plan to address the generalizability of the approach?
> >
> > Please see the answer to Weakness 3.
> >
> > - **Question 3**
> >
> > > The additional data generated for Active Learning are selected based only on errors. Without some sort of approach to balance the introduction of new data, does the approach bias the dataset distribution causing the learned distribution to fail to generalize to other molecules?
> >
> > First of all, we would like to note that we test the NNPs on a subset from the same distribution than the train dataset. The train dataset $\mathcal{D}\_0$ and the evaluation dataset $\mathcal{D}\_{\text{test}}$ share no molecules. Our evaluation results described in Section 6.1 and 6.2 show that the NNP successfully generalizes to unseen molecules from the same distribution. Considering the balancing of the new data, we use the following technique: 10% of each mini-batch is sampled from the initial training dataset $\mathcal{D}\_0$. We have added this detail to the appendix of the revised manuscript.
> >
> > - **Question 4**
> >
> > > How different is the GOLF approach from the active learning technique presented by Kulichecnko et al. (2023).
> >
> > The method presented in the Kulichenko et al. paper is an interesting and promising approach for the exploration of the configuration space of a single molecule. Unfortunately, it requires training a new NNP ensemble for each new molecule, which is a completely different problem setup compared with ours. Moreover, the focus of the method is to enrich the MD trajectories with the conformations for which the ensemble is uncertain, which is shown to improve the energy prediction. In theory, this approach may also improve the atomic forces prediction, but it remains to be studied.

---

### Official Review · Reviewer_bemc · 2023-11-01

**Soundness:** 2 fair
**Presentation:** 2 fair
**Contribution:** 2 fair
**Rating:** 6
**Confidence:** 4

**Summary:**

This paper proposes a gradual optimization learning framework (GOLF) for molecular conformation minimization. The framework is designed to improve the training of Neural Network Potentials (NNP). The authors first claim that NNPs trained on existing datasets suboptimal in energy minimization due to the distribution shift and perform experiments to show (a large amount of) additional data from the optimization trajectories can help improve the NNP's performance.  The GOLF framework uses a surrogate oracle (MMFF) to evaluate the conformation energy and expand the training data by selecting the incorrect prediction and re-evaluating with the genuine oracle (DFT), which reduces the required additional data. The experiments on the nablaDFT dataset demonstrate the effectiveness of the proposed method.

**Strengths:**

The proposed method sounds reasonable and the experiments shows its effectiveness.  The method also looks easy to implement, which can improve the conformation energy minimization performance at a small cost.

**Weaknesses:**

- The writing is not very clear, especially in the introduction section. It takes me a while to understand simply enriching the training dataset is actually a preliminary baseline method the authors want to compare with. Also, lots of experiment details are mixed with the method, which makes the paper not very easy to read.
- The calculation of COV and MAT looks problematic. It seems the authors optimize **one conformation per molecule** and take them of the entire test set as the generation set. However, in the conformation generation setting, models generate **multiple conformations per molecule** to construct the generation set, and then COV and MAT are calculated per molecule,  and finally the average / median of them on the entire test set is reported.

**Questions:**

- Are the conformations in nablaDFT dataset equilibrium ones or the intermediate state sampled from the optimization process? How large is the training set D0? More description about this dataset is needed.
- ConfOpt and TorsionDiff are designed to generate equilibrium low-energy conformers, and not guaranteed to achieve a lower energy by repeatedly applied. Thus, I think it's unfair to compare these models with GOLF in terms of pct.
- The statement of "We hypothesize that in the case of ConfOpt, the main problems are the choice of the architecture and the fact that the model generates optimal conformations from SMILES and does not use initial geometries. " doesn't make sense to me. ConfOpt takes the 2D molecular graph as input and also utilizes initial 3D conformations.
- Does $f^{traj-10k /100k }$ keep the total number of updates equal to $5 \times 10^5$? If so, please provide more training details, otherwise, the comparison between them and $f^{GOLF}$ is unfair.

---

> ### Author Response · Authors · 2023-11-23
> **Answer to Official Review of Submission9224 by Reviewer bemc (1/2)**
>
> ## Weaknesses
>
> - **Weakness 1**
>
> > The writing is not very clear, especially in the introduction section. It takes me a while to understand simply enriching the training dataset is actually a preliminary baseline method the authors want to compare with. Also, lots of experiment details are mixed with the method, which makes the paper not very easy to read.
>
> We agree that some additional context regarding the use of $f^{\text{traj-*}}$ as a baseline is needed. We added clarifying sentences in the revision of the manuscript. Unfortunately, we do not see a way to separate the method from some experiment details, as the GOLF is mainly motivated by the empirical results obtained in section 4. We believe that introducing our framework without a preliminary discussion of experimental details and results will be more confusing for the reader. We will be happy to consider any suggestions on improving the writing of the manuscript.
>
> - **Weakness 2**
>
> > The calculation of COV and MAT looks problematic. It seems the authors optimize one conformation per molecule and take them of the entire test set as the generation set. However, in the conformation generation setting, models generate multiple conformations per molecule to construct the generation set, and then COV and MAT are calculated per molecule, and finally the average / median of them on the entire test set is reported.
>
> We agree that such metrics are indeed not ideal for the conformation optimization task due to the problem you described above. However, we decided to compute these metrics to provide a fair comparison with ConfOpt and Torsional Diffusion, as the main results in these papers are reported in terms of COV and MAT.
>
> ## Questions
>
> - **Question 1**
>
> > Are the conformations in nablaDFT dataset equilibrium ones or the intermediate state sampled from the optimization process? How large is the training set D0? More description about this dataset is needed.
>
> The conformations in the nablaDFT are generated using the following procedure:
>
> 1. First, several thousand different conformations per molecule are sampled using Rdkit's ETKG [1] based on a SMILES representation of the molecule.
> 2. Second, these conformations are clustered using Butina [2].
> 3. The authors then identify the smallest number of clusters needed to cover 95% of generated conformations.
> 4. Centroids of these identified clusters are chosen and evaluated with a DFT-based oracle.
>
> The training dataset contains 4000 molecules and 10000 conformations (~2.5 conformations per molecule). We apologize for not providing this information in the first version of the paper. We added this information in the revision. We also added a brief description of the nablaDFT dataset in the appendix of the revised manuscript.
>
> - **Question 2**
>
> > ConfOpt and TorsionDiff are designed to generate equilibrium low-energy conformers, and not guaranteed to achieve a lower energy by repeatedly applied. Thus, I think it's unfair to compare these models with GOLF in terms of pct.
>
> We would like to clarify that in our experiments, these models are trained to generate a single optimal conformation from an initial random conformation. We use these models in the intended way and do not repeatedly apply any of them. The training dataset for these models consists of optimal conformations obtained with $\mathcal{O}_G$ and, thus, we expect them to generate optimal conformations, altough the inference of these models is different to $f^{\text{GOLF-*}}$, we think it is still valuable to compare the generative approach (ConfOpt, TorsionDiff) with an iterative optimization approach.
>
> - **Question 3**
>
> > The statement of "We hypothesize that in the case of ConfOpt, the main problems are the choice of the architecture and the fact that the model generates optimal conformations from SMILES and does not use initial geometries. " doesn't make sense to me. ConfOpt takes the 2D molecular graph as input and also utilizes initial 3D conformations.
>
> We follow the original implementation [https://github.com/guanjq/confopt_official], which takes the 2D molecular graph, generates a random conformation with Rdkit’s ETKG, and optimizes it with MMFF. This implementation does not provide any means to supply the model with an initial conformation of our choice.
>
> [1] [Wang, S., Witek, J., Landrum, G. A., & Riniker, S. (2020). Improving conformer generation for small rings and macrocycles based on distance geometry and experimental torsional-angle preferences. *Journal of chemical information and modeling*, *60*(4), 2044-2058.]
>
> [2] Barnard, J. M., & Downs, G. M. (1992). Clustering of chemical structures based on two-dimensional similarity measures. *Journal of chemical information and computer sciences*, *32*(6), 644-649.

---

> > ### Author Response · Authors · 2023-11-23
> > **Answer to Official Review of Submission9224 by Reviewer bemc (2/2)**
> >
> > - **Question 4**
> >
> > > Does $f^{\text{traj-10k/100k}}$ keep the total number of updates equal to $5 \times 10^5$? If so, please provide more training details, otherwise, the comparison between them and is unfair.
> >
> > Yes, the number of NN updates for all models except for $f^{\text{traj-500k}}$ is equal to $5 \times 10^5$. For $f^{\text{traj-500k}}$ the number of updates is equal to $10^6$ as $5 \times 10^5$ is not enough for the training to converge. We have added a table with hyperparameters along with an additional explanatory sentence in the appendix.

---

### Official Review · Reviewer_DBvP · 2023-11-05

**Soundness:** 3 good
**Presentation:** 3 good
**Contribution:** 2 fair
**Rating:** 6
**Confidence:** 3

**Summary:**

This paper introduces GOLF, a framework for improving molecular conformation optimization with neural networks. GOLF addresses distribution shift issues, enhancing energy prediction and optimization. It outperforms traditional methods and reduces the need for physical simulator interactions by 50 times.

**Strengths:**

1. The development of the proposed GOLF is clear.
2. Demonstrated outstanding performance in conformation optimization tasks.
3. Generalization to larger molecules.

**Weaknesses:**

1. Dataset Limitation: The paper may be limited by the availability and diversity of datasets used for testing, potentially impacting the generalizability of the results.
2. Complexity: It seems that the complexity of GOLF is not clearly discussed in the paper.
3. Practical Implementation: Though the algorithm is not very complicated, this paper does not release code, which leaves me cautious about the practical implementation and complexity of the algorithm.

Overall, while the paper presents valuable contributions, addressing these weaknesses could enhance its overall impact and relevance in the field of molecular conformation optimization.

**Questions:**

Same with the 'weaknesses' part:
1. Why not use more datasets besides nablaDFT?
2. What's the complexity of GOLF? It seems that you only show experiment results on subsets of nablaDFT. Is it because the computational complexity of the method is very high?

---

> ### Author Response · Authors · 2023-11-23
> **Answer to Official Review of Submission9224 by Reviewer DBvP (1/ 2)**
>
> ## Weaknesses
>
> - **Weakness 1**
>
> > Dataset Limitation: The paper may be limited by the availability and diversity of datasets used for testing, potentially impacting the generalizability of the results.
>
> We agree that the experimental section could benefit from a more diverse evaluation dataset. Therefore we significantly extended the main evaluation dataset (10x) by sampling more conformations with less than 35 atoms from the nablaDFT dataset. The extended evaluation dataset contains 20k conformations for X molecules. It contains  $\mathcal{D}\_{\text{test}}$ as its subset.
> Moreover, we selected 1828 molecules (1 conformation per molecule) from the SPICE dataset, optimized them with $\mathcal{O}\_G$ to get the optimal energy. We evaluated all non-generative models from our paper on this new SPICE test set $\mathcal{D}\_{\text{test}}^{\text{SPICE}}$.
> We provide the results in the table below. We understand that the size of the $\mathcal{D}\_{\text{test}}^{\text{SPICE}}$ may raise similar concerns, but we were unable to collect a larger dataset during the rebuttal due to the need for an expensive $\mathcal{O}\_G$ optimizations. We will extend the $\mathcal{D}\_{\text{test}}^{\text{SPICE}}$ to ~20k conformations and provide results for generative models in the camera-ready version.
>
> ### SPICE test (1828 conformations)
> |  | OpenFF | RDKit | $f^{baseline}$ | $f^{traj-10k}$ | $f^{traj-100k}$ | $f^{traj-500k}$ | $f^{GOLF-1k}$ | $f^{GOLF-10k}$ | $SPICE-f^{baseline}$ | $SPICE-f^{GOLF-10k}$ |
> | --- | --- | --- | --- | --- | --- | --- | --- | --- | --- | --- |
> | $\overline{\operatorname{pct}}\_{100}$ | 61.14 | 77.58 | 72.71 | 73.22 | 77.06 | 80.25 | 80.31 | 82.41 | 79.76 | 86.59 |
> | $\operatorname{pct}\_{\text{div}}$ | 5.38 | 8.15 | 27.73 | 17.45 | 16.02 | 12.80 | 13.56 | 12.41 | 23.08 | 15.09 |
> | $\overline{E^{\text{res}}\_{100}}$ | 12.58 | 8.42 | 13.62 | 9.63 | 8.09 | 6.88 | 6.63 | 6.06 | 6.06 | 4.19 |
> | $\operatorname{pct}\_{\text{solved}}$ | 0.05 | 10.72 | 6.35 | 5.17 | 15.90 | 25.65 | 17.53 | 27.73 | 8.91 | 19.41 |
>
> The OpenFF is a non-neural force field used in the original SPICE paper [1]. $SPICE-f^{baseline}$ was trained on ~10000 conformations (2.5 conformations per molecule) from the SPICE dataset. $SPICE-f^{GOLF-10k}$ was trained with the same hyperparameters as $f^{GOLF-10k}$ and used $SPICE-f^{baseline}$ to initialize the NNP. These preliminary results show that models trained on nablaDFT and SPICE datasets demonstrate comparable performance on $\mathcal{D}\_{\text{test}}^{\text{SPICE}}$.
>
> ### Extended test (19477 conformations)
>
> |  | RDKit | $f^{baseline}$ | $f^{rdkit}$ | $f^{traj-10k}$ | $f^{traj-100k}$ | $f^{traj-500k}$ | $f^{GOLF-10k}$ |
> | --- | --- | --- | --- | --- | --- | --- | --- |
> | $\overline{\operatorname{pct}}\_{100}$ | 85.47 | 77.88 | 93.04 | 95.08 | 96.15 | 98.75 | 98.78 |
> | $\operatorname{pct}\_{\text{div}}$ | 0.62 | 7.46 | 4.43 | 4.50 | 2.77 | 1.99 | 2.98 |
> | $\overline{E^{\text{res}}\_{100}}$ | 5.50 | 8.58 | 2.77 | 1.96 | 1.50 | 0.53 | 0.51 |
> |  $\operatorname{pct}\_{\text{solved}}$ | 4.05 | 8.18 | 35.42 | 37.01 | 52.71 | 73.41 | 77.26 |
>
> These results show that all conclusions made in the first version of the paper hold on an extended evaluation dataset.
>
> - **Weakness 2**
>
> > Complexity: It seems that the complexity of GOLF is not clearly discussed in the paper.
>
> Inference: GOLF uses NNPs that rely on Message Passing [2]. In the worst case, the computational complexity is quadratic in terms of the number of atoms in the molecule. We provide average inference time for baselines and NNPs and compare them with the average $\mathcal{O}_G$. All the measurements were made on the same machine with a 16-core Intel(R) Xeon(R) Gold 6278C and a single Tesla V100 GPU. The results are in the table below.
>
> | Model | Iterative Optimization with NNP (100 steps) | ConfOpt (single step) | Torsional Diffusion (single step) | UniMol+ (single step) | Iterative Optimization with $\mathcal{O}_G$ (until convergence, ~25 steps on average) |
> | --- | --- | --- | --- | --- | --- |
> | Average Inference time (seconds) | 1.312 | 0.005 | 0.48 | 0.002 | 2634.787 |
>
> Training: around 2/3 of the computational complexity during training comes from various NN operations. We use DFT-based oracle, which complexity is $O(N^4)$ [3]. For example, the training of the GOLF-10k on our machine with 120 workers for DFT calculations takes approximately 20 hours .
>
> [1] Eastman, P., Behara, P. K., Dotson, D. L., Galvelis, R., Herr, J. E., Horton, J. T., ... & Markland, T. E. (2023). Spice, a dataset of drug-like molecules and peptides for training machine learning potentials. Scientific Data, 10(1), 11.
>
> [2] Gilmer, J., Schoenholz, S. S., Riley, P. F., Vinyals, O., & Dahl, G. E. (2020). Message passing neural networks. *Machine learning meets quantum physics*, 199-214.
>
> [3] Kohn, W., & Sham, L. J. (1965). Self-consistent equations including exchange and correlation effects. *Physical review*, *140*(4A), A1133.

---

> ### Author Response · Authors · 2023-11-23
> **Answer to Official Review of Submission9224 by Reviewer DBvP (2/ 2)**
>
> ## Questions
>
> - **Question 1**
>
> > Why not use more datasets besides nablaDFT?
>
> We have not considered datasets such as QM9 and GEOM as they contain equilibrium conformations. We have not considered MD17 and other similar datasets due to the low diversity of the molecules.
>
> Additionally, to answer your first question, we trained the $f^{\text{baseline}}$ and the $f^{\text{GOLF-10k}}$ on a subset of SPICE dataset. For the SPICE training set $\mathcal{D}^{\text{SPICE}}\_{\text{train}}$, we collected 4000 molecules with roughly 2.5 conformations per molecule resulting in approximately 10k conformations (the same size as $\mathcal{D}\_{\text{train}}$). We evaluated these models both on the $\mathcal{D}\_{\text{test}}^{\text{SPICE}}$ and the extended $\mathcal{D}\_{\text{test}}$. The results are in the table below.
>
> | NablaDFT | $f^{GOLF-10k}$ | $\operatorname{SPICE}-f^{GOLF-10k}$ |
> | --- | --- | --- |
> | $\overline{\operatorname{pct}}_{100}$ | 98.78 | 91.71 |
> | $\operatorname{pct}_{\text{div}}$ | 2.98 | 12.69 |
> | $\overline{E^{\text{res}}\_{100}}$ | 0.51 | 3.27 |
> | $\operatorname{pct}\_{\text{solved}}$ | 77.26 | 17.27 |
>
> | SPICE | $f^{GOLF-10k}$ | $\operatorname{SPICE}-f^{GOLF-10k}$ |
> | --- | --- | --- |
> | $\overline{\operatorname{pct}}_{100}$ | 82.41 | 86.59 |
> | $\operatorname{pct}_{\text{div}}$ | 12.41 | 15.09 |
> | $\overline{E^{\text{res}}\_{100}}$ | 6.06 | 4.19 |
> | $\operatorname{pct}\_{\text{solved}}$ | 27.73 | 19.41 |
>
> The main take-away from these tables is that the percentage of optimizations solved is larger for the $f^{\text{GOLF-10k}}$ trained on NablaDFT. This is likely due to the presence of near-optimal conformations in SPICE which can in theory result in collecting highly correlated training samples.
>
> - **Question 2**
>
> > What's the complexity of GOLF? It seems that you only show experiment results on subsets of nablaDFT. Is it because the computational complexity of the method is very high?
>
> The high computational complexity of the evaluation procedure is tied to the need to optimize every conformation with $\mathcal{O}_G$. Estimating the percentage of optimized energy and other reported metrics requires running a minimization procedure with a DFT-based oracle. For example, it takes ~593 CPU days to calculate optimal energies for 19477 conformations in the extended evaluation dataset. While the optimization of 19477 conformation with GOLF-10k takes ~7.1 hours on a single V100 GPU.

---

### Author Response · Authors · 2023-11-23
**General answer**

We thank all the reviewers for pointing out the flaws of the manuscript. This helped us to significantly improve the writing and the presentation of the paper. We list the main changes below and we will refer to them in personal answers:

1. Reworked and fixed “Notaion and Preliminaries” (including the renaming of GO and SO to $\mathcal{O}_G, \mathcal{O}_S$).
2. Moved “External Optimizers” experiment to Appendix: results, yet interesting, are not of high importance for the paper.
3. Added metrics. We would like to thank the reviewer RbRE for pointing out that it is important to explain which percentage of optimized energy may be treated as the “solution”. We added the explanation of the *chemical precision* to the manuscript and provided additional energy metrics for all the models along with the percentage of “solved” conformations (we call it $\operatorname{pct}\_{\text{success}}$), which is a percentage of final conformations with energy less than optimal plus *chemical precision*.
4. Extended the evaluation of the proposed approach: a larger nablaDFT subset (19477 conformations) and a subset of SPICE (1832 conformations). We will extend the SPICE evaluation dataset in the camera-ready version.
5. Fixed Table 3 in Section 6.4 (”Larger Molecules”). The DFT computation timeout was initially chosen for smaller molecular size, we have found that increasing this parameter reduces the percentage of diverged optimizations.
6. Improved writing and presentation of the manuscript.
7. Released the source code.
8. Extended the appendix:
    1. Added the hyperparameter tables and training details that were missing in the initial submit.
    2. Added the description of nablaDFT dataset and the way the conformations are collected.
    3. Added a visualization of final conformations for different optimization methods.

---

### Meta-Review · Area_Chair_VMac · 2023-12-08

**Metareview:**

The paper first studied the out-of-distribution generalization challenge when using a neural network potential (NNP) which shows significantly more data than large static datasets are required for successful geometry optimization, then proposes the GOLF method, an active learning strategy that achieves successful geometry optimization with much fewer additional data. The empirical investigation is done in fine quality, and the empirical advantage seems significant. Generalization to larger-scale molecules is also demonstrated. Nevertheless, the proposed method does not seem particularly technically inspiring and requires a third predictor (in addition to the NNP and the accurate labeler). It also lacks comparison to other active learning methods (even in a smaller scale).

**Justification For Why Not Higher Score:**

There remain weaknesses as mentioned in the Metareview.

**Justification For Why Not Lower Score:**

The paper makes a valuable contribution to highlight the generalization issue of NNP in real use cases to the community, and proposes an effective method that addresses the problem to a certain promising extent. The authors seem to have addressed other issues raised by the reviewers.

---

### Decision · Program_Chairs · 2024-01-16

Accept (poster)